## Registered report

psychology/behaviour/cognition

repeated economic games, self-reported questionnaires, variability, external validity, social trust, trust game

**Author for correspondence:**
L. Safra
e-mail: lou.safra@sciencespo.fr

[25][†]These authors contributed equally to this study.

# Variability in repeated economic games: comparing trust game decisions to other social trust measures

L. Safra[1,2], N. Lettinga[1], P. O. Jacquet[1,3,4,†] and C. Chevallier[1,†]

[1]LNC, Département d'études cognitives, Ecole normale supérieure, Université PSL, INSERM, 75005 Paris, France
[2]Sciences Po, CEVIPOF, CNRS, Paris, France
[3]Centre de rercherche en Epidémiologie et Santé des Populations, Université Paris-Saclay, Université Versailles Saint-Quentin, 94807 Villejuif, France
[4]Institut du Psychotraumatisme de l'Enfant et de l'Adolescent, Centre Hospitalier de Versailles et Conseil départemental des Yvelines et des Hauts de Seine, Versailles 78000, France

(iD) LS, 0000-0001-7618-6735; NL, 0000-0001-7173-3955

Economic games are well-established tools that offer a convenient approach to study social behaviour. Although widely used, recent evidence suggests that decisions made in the context of standard economic games are less predictive of real-world behaviour than previously assumed. A possible explanation for this discrepancy is that economic games decisions in the laboratory are more likely to be influenced by the current situation, while questionnaires are specifically designed to measure people's average behaviour across a long period of time. To test this hypothesis, we performed a longitudinal study where 275 respondents played 16 Trust games every two days within a three-week period, and filled out a questionnaire that measures social trust. This study confirmed the instability of our measure of trust behaviour over time and the substantial stability of questionnaire responses. However, we found a significant association between self-reported social trust and participants' average behaviour in the trust game measured across sessions, but also with participants' behaviour measured only in Session 1. Nevertheless, analysis of behavioural changes in the Trust games over time revealed different behavioural profiles, highlighting how economic games and questionnaires can complement each other in the study of social trust.

# 1. Introduction

Economic games are well-established tools across different scientific disciplines such as economics and the social sciences [1–3] and offer a convenient approach to study social behaviour (e.g. trust). Economic games have become a pillar of experimental research with literally thousands of studies using them [4]. Although economic games are widely used, recent evidence suggests that economic games decisions are less predictive of real-world behaviour than previously assumed [5–7]. Most notably, Galizzi & Navarro-Martinez [4] conducted a systematic review and meta-analysis, and found a weak correlation ($r = 0.14$) between economic games and social behaviour in the field, indicating low external validity. They also performed an original, large and comprehensive laboratory-field experiment, where the same sample of participants played several economic games in the laboratory and were confronted with naturalistic situations related to social preferences in the field. Their experiments show that economic games do a poor job at explaining social behaviours in the field ($r_s = 0.03$).

Galizzi & Navarro-Martinez's [4] work focused on the Dictator game, the Ultimatum game, the Public Goods game and the Trust game, and it is possible that other economic games are more predictive of real-world behaviour. Nonetheless, the four economic games analysed by Galizzi & Navarro-Martinez [4] cover a large fraction of experimental research on social preferences. In what follows, we mostly refer to evidence pointing to a lack of external validity of these standard four economic games.

The reasons why economic games are less predictive of real-world behaviour remain unclear but one possible explanation is that economic games decisions are not very stable over time. Economic games are supposed to measure preferences, which are assumed to be stable [8], but test–retest reliability is in fact low. For example, using a design where participants played the same games over a three-month period (i.e. various Dictator and Prisoner's Dilemma games), (Brosig *et al.* 2007, unpublished manuscript) found that stability was low, while Lönnqvist *et al.* [9] found that decisions in the Trust game were stable over a 1-year period. Unfortunately, both of these studies relied on very small samples ($N = 40$ and $N = 22$ for Brosig *et al.* 2007, unpublished manuscript and Lönnqvist *et al.* [9], respectively), which considerably limits the generalizability of the results. Importantly, temporal instability of decisions made in other behavioural domains, such as risk and time preferences, has been found in other studies including large samples (see review in [10,11]). For example, Frey *et al.* [11], using a battery of behavioural risk tasks, found that they were less correlated among themselves over a six-month period.

Overall, self-reported questionnaires seem to be more stable and more predictive of real-world behaviour [9,11,12]. Although, self-reported questionnaires have their own limitations: people can be biased by social desirability [13] and sometimes have limited insight on their mental states and preferences [14]. Nonetheless, such a misalignment between self-reported questionnaires and laboratory-based experimental tasks which include economic games are frequently found in the literature [4,9–12,15–23]. This raises the interesting question of why economic games and self-reported questionnaires, which are supposed to measure the same construct, provide such a different perspective.

A possible explanation for the instability of laboratory-based experimental tasks such as economic games, is that decisions in the laboratory are likely influenced by the current situation (i.e. people's current states) where the participant is in, while many self-reported questionnaires are specifically designed to capture people's average behaviour across a long period of time (i.e. people's stable traits) [24]. Some questionnaires aim to measure participants' states or both their current state and their stable traits (e.g. the State-trait anxiety questionnaire, [25]). In the specific case, we are interested in, social trust questionnaires are designed to measure an individual's prototypical tendency to trust other people. For example, questionnaire items are often phrased as: 'Generally speaking…'. Overall, such design features are geared to reduce the influence of momentary trends [26,27]. By contrast, economic games are performed in specific and highly structured situations, which give more weight to the current state the participant is in, and thus increase variability in responses [18,28]. For instance, studies have found that economic game decisions are affected by the emotional state of the participant, noise and illumination, distance to screen, the presence of other people, being under scrutiny of the experimenter, non-anonymity, and artificial stakes, choice sets and time horizons [6,18].

If the above explanation is correct, then we should see that repeating economic games over an extended period of time should bring participants closer to the average behaviour that questionnaires capture. The reasoning is that by performing repeated economic games, the influence of any momentary, within-person fluctuations (i.e. participants' states) should be counteracted. The goal of this paper is therefore to investigate whether economic game decisions made over an extended period of time approximate questionnaires responses.

To test this hypothesis, we focus on people's social trust. Social trust is usually conceptualized along two dimensions: trust propensity displayed by the trustor and trustworthiness displayed by the trustee [29]. In our study, social trust refers to the trustors' propensity to trust others. Trust propensity varies considerably between countries (e.g. [30,31]) and individuals (e.g. [32–34]). However, an individual's trust propensity is typically conceptualized as a stable trait [29,35–37]. For instance, Mayer *et al.* [29] write that trust propensity is 'a trait that is stable across situations'. Furthermore, there are different types of trust (e.g. in family, in close friends, in institutions, in unknown people generally, etc.). Each type may have different psychological underpinnings and may be sensitive to different factors. Here we are particularly interested in generalized social trust [38], which is trust in unknown individuals.

We chose to focus on social trust because the economic game (i.e. the Trust game, see [39,40]) and questionnaire [22] that are frequently used to measure it are each well-established tools that are used in the literature as standard. Both of these measures try to capture people's generalized social trust. However, recent data demonstrate that participants' decisions in the Trust game and their answers to social trust questionnaires are not correlated [4,41,42]. Our goal is to test the following hypotheses: (1) decisions in the Trust game show substantial variability over time (i.e. low test–retest reliability); (2) social trust questionnaire responses show substantial stability over time (i.e. high test–retest reliability); (3) the average behaviour in the Trust game on the first session is not or poorly correlated with social trust questionnaire responses while the average behaviour in the Trust game over all sessions is significantly and more strongly correlated with social trust questionnaire responses.

# 2. Material and methods

## 2.1. Preregistration: respondents

Our sample will consist of 275 respondents recruited via the online platform Prolific Academic (see §2.7 'Power analysis and sample-size estimation' for details). We will use Prolific's pre-screening criteria to filter respondents by age, nationality, language, approval rate and number of previous submissions. We will only recruit respondents older than 18 years, who are from the UK and speak English as their first language, and who have a minimum approval rate of 90% and more than 10 previous submissions on Prolific. Respondents will be excluded when they do not complete the entire experiment, when they do not provide correct answers during the last of the two comprehension check sections on the first and last session of the experiment, when their reaction time is below 200 ms on at least 90% of the games, or when multiple respondents use the same IP address. The recruitment of respondents will be stopped once our target sample is reached on the first session of the experiment.

## 2.2. Preregistration: overall design and procedure

Respondents will be asked to play several Trust games over an extended period of time, making it a longitudinal study. The respondents will play the Trust games on 10 sessions taking place every other days over a three-week period. The total time to complete the study is around 1 h. Respondents will receive their entire compensation at the end of the three-week study. The experiment is programmed on Qualtrics. The experimental protocol is approved by the local Ethical Committee (Conseil d'évaluation éthique pour les recherches en santé - CERES no. 201659).

During the first session, respondents will sign an informed consent form, play the Trust game 16 times and fill out the self-reported trust questionnaire, in that order. For the Trust game, detailed instructions and two comprehension check sections are included beforehand (the full questionnaire for session 1 is provided in electronic supplementary material S1). In the eight intermediate sessions, respondents play the Trust game 16 times (detailed instructions and comprehension check sections are again included). During the last session, respondents play the Trust game 16 times again and fill out the self-reported trust questionnaire. After that, respondents are confronted with two scenarios where they can display real-world behaviour related to social trust: data sharing and actual helping. We included these two measures for exploratory purposes. The goal is to see whether people's Trust game decisions and responses to the questionnaire are correlated with actual behaviour. For data sharing, respondents have the option to give permission to share their data from 13 additional questions about their personal life with other researchers. For actual helping, respondents have the option to help the researchers by providing feedback on instructions for a future experiment. At the

end of the experiment, respondents are paid their participation fee (£7.5) and their total earnings based on one of the Trust games per session that will be chosen randomly (expected average = £5.75).

## 2.3 Preregistration: variables of interest

### 2.3.1. Trust game

For the Trust game [39], we adopted a similar methodology as Mell *et al*. [34]. A total of 16 independent games will be played per session. The respondent always has the role of trustor and will play with 16 different trustees. At the start of each game, both players receive five tokens. Respondents will be informed that the tokens earned by the end of the experiment will be transformed into a real monetary bonus (each token is worth £0.1). The amount of tokens that the trustor can give (1, 2, 4 or 5) and the probability of reciprocation by the trustee (0.65, 0.70, 0.75 or 0.80%) are manipulated per game. Each combination will be played once per session, resulting in the 16 different games (based on the four different stakes and four reciprocation probabilities), and will be provided in random order. For each game, respondents will be presented with how many tokens are required to play the game (i.e. what they have to give in order to play this particular game) and the probability of reciprocation by this particular trustee (an example of the screen that respondents are presented with is provided in electronic supplementary material S2). For each game, the respondents' willingness-to-play is the dependent variable, and will be measured by the question: 'How much do you want to play with this partner?' (scale 1–9; from 'not at all' to 'extremely'), with higher scores indicating higher levels of social trust. Playing the game means transferring the amount of tokens that is given. The amount given is multiplied by three. The trustee can then choose to keep all the tokens the participant gave or to share some of the tokens he or she now has. If the trustee chooses to share, then both players end up with the same number of tokens.

The respondents will be told that they will play with 'virtual partners', and that each game is played with a different virtual partner (this counteracts any reputation effects) who behaves independently. Respondents will be informed that these virtual partners are not real people but are programmes that can simulate real-life behaviour, just like characters in videogames. Importantly, recent work shows that people invest similarly in humans and robots in a Trust game where the participants knew their partner was a robot or human [43]. In order to avoid learning effects, the response of the trustee is not shown. Instead, one game per session will be randomly chosen to calculate the bonus that will be revealed at the end of each session and added to the respondents' payment at the end of the last session of the experiment. Calculating the bonus for each session is done in two steps. First, one of the 16 Trust games is randomly chosen. Second, whether or not this randomly selected game is played will be calculated. This is done by randomly drawing a number between 1 and 9; if this number is below the respondents' willingness-to-play, the game is played and the trustee's decision to reciprocate is covertly simulated based on her reciprocation probability. If the number is above or equal to the respondents' willingness-to-play, the game is not played and the respondent keeps his/her five tokens.

### 2.3.2. Self-reported social trust

Self-reported social trust will be measured on the first and last session of the experiment with three questions from the European Social Survey [22]. The questions are 'Generally speaking, would you say that most people can be trusted, or that you can't be too careful in dealing with people?', 'Do you think that most people would try to take advantage of you if they got the chance, or would they try to be fair?' and 'Would you say that most of the time people try to be helpful or that they are mostly looking out for themselves?' (scale 0–10), with higher scores indicating higher levels of social trust. For each question, respondents have the option to reply 'I don't know', in which case their answer will be recoded as missing. Answers to the three items will be *z*-scored and averaged into a single index.

### 2.3.3. Data sharing

A new scenario proposed by Bauer *et al*. [44] to measure real-world behaviour related to social trust is whether respondents share, i.e. entrust their data to others. Specifically, they ask respondents for their permission to include additional external private data (i.e. data from social insurance carriers) to be included in the analysis. Respondents are therefore asked to trust the researchers with this additional data and to not abuse it, making it straightforwardly related to social trust. Here we adopt a similar

strategy. At the end of the experiment on session 10 (after the Trust games are played and the three trust items are filled out), all respondents will be presented with a set of 13 additional questions about their personal life. This questionnaire will be entitled 'Personal information about your past and current life', and will be introduced as follows:

> In the final part of this experiment, you will be presented with 13 additional questions about your personal life. These questions relate to the way your life was going on at home during childhood, as well as your current life, such as your financial situation and your health status.

Items 1 and 2 inform about the respondents gender (item 1) and age in years (item 2). Items 3–8 are taken from the well-established questionnaire designed by Mittal *et al.* [45], and which assess the life conditions that respondents experienced during their childhood. These items inform about the socio-economic status of the family household and the unpredictability of the family environment of the respondents when they were younger than 10 years of age. The following instructions will first be presented: 'Think back to your life when you were younger than ten. This time includes preschool, kindergarten, and the first few years of elementary school'. Then respondents will be asked to say how much they agree with the following six statements: 'When I was younger than 10… : 'My family usually had enough money for things when I was growing up', 'I grew up in a relatively wealthy neighborhood', 'I felt relatively wealthy compared to the other kids in my school', 'things were often chaotic in my house', 'people often moved in and out of my house on a pretty random basis', and 'I had a hard time knowing what my parent(s) or other people in my house were going to say or do from day-to-day.' Responses are made on 7-point scales ranging from 1: strongly disagree, to 7: strongly agree. Items 9–11 inform about the respondents' socio-economic status at the time of the inclusion in the experimental protocol. They are taken from the well-validated work by Griskevicius *et al.* [46] and ask respondents to say how much they agree with the following three statements: 'I have enough money to buy things I want', 'I don't need to worry too much about paying my bills', and 'I don't think I'll have to worry about money too much in the future'. Finally, items 12 and 13 inform about the respondents' health state. More specifically, item 12 asks the question 'How is your health in general?', and proposes four responses from 1-'Bad' to 4-'Excellent'. Item 13 asks the respondents to answer—by choosing a value between 0 and 100—the question 'How much effort do you make to look after your health and ensure your safety these days?'

Right after the questionnaire, the respondents are shown the following scenario:

> You have just been presented with 13 additional questions about your personal life. If you agree, we would like to share your answers to these extra questions and link them with the data collected during the past 10 sessions. We would like to ask you to give your consent for this extra data to be linked to the survey data and shared with other researchers. All data protection regulations will be strictly observed during the process, i.e., the results will be anonymous and will not allow any conclusions to be drawn about your person. Your consent is of course voluntary. You can also revoke it at any time. Do you agree to the extra data being linked and shared with other researchers?

Respondents can either select 'yes' or 'no' to this last question. Only if respondents select 'yes', will their extra data be linked to the survey data and shared online on the Open Science Framework. We expect that the average behaviour in the Trust game during the first session is not or poorly correlated with data sharing, but social trust questionnaire responses and the average behaviour in the Trust game over all sessions are significantly and more strongly correlated with data sharing.

### 2.3.4. Actual helping

Another scenario to measure real-world behaviour related to social trust is whether respondents display actual helping behaviour by providing feedback on instructions for a future experiment that the researchers are developing [47]. Respondents are explicitly told that this is optional, without compensation, and will not affect their earnings from the current experiment. We adopted a similar scenario to Peysakhovich *et al.* [47] (answer categories: yes/no), which is provided in electronic supplementary material, S3. Respondents who choose to help the researchers are shown a set of instructions and are asked for comments. Again, we expect that the average behaviour in the Trust game during the first session is not or poorly correlated with actual helping, but social trust questionnaire responses and the average behaviour in the Trust game over all sessions are significantly and more strongly correlated with actual helping.

## 2.4. Preregistration: positive control checks

To determine if our sample is representative in terms of the average willingness-to-play in the Trust games on the first session, our data will be compared to Mell *et al.* [34], who used a similar methodology and found an average value of 6.02 (s.d. = 1.49) in their initial study and 6.03 (s.d. = 1.33) in a replication study.

We expect that our mean is close to the means found by Mell *et al.* [34], specifically between 4.62 and 7.44, because this is exactly one standard deviation below and above the mean found by Mell *et al.* [34].

## 2.5. Preregistration: data quality check and data cleaning

The assumption of normal distribution will be inspected for the average willingness-to-play in the Trust games (for the first session and all sessions combined) and the social trust questionnaire responses (for the first session and last session) using *P-P* plots and the Shapiro–Wilk test. An inter-item reliability analysis (i.e. Cronbach's alpha) will be performed on the three self-reported social trust questions for the first and last session of the experiment separately.

## 2.6. Preregistration: pre-registered analyses

All analyses will be carried out in R 4.0.4 (https://www.r-project.org/) with R Studio 1.2.5042. The data consist of a within-subject, repeated measures design because the study is longitudinal in nature.

To test the hypothesis that decisions in the Trust game show substantial variability over time, we will measure the test–retest reliability of the willingness-to-play via the intraclass correlation coefficient (ICC), which is a widely used reliability index. To determine the ICC, we will first fit an ordinal logistic regression model using the function *clmm2* from the R package *Ordinal*. The dependent variable is willingness-to-play, stakes and reciprocation probability will be included as fixed effects and respondents' ID will be included as a grouping factor with a random intercept. After that, we will calculate the ICC using the following formula: $ICC = \tau/(\tau + (\pi^2))$, where $\tau$ is the between group variance, in our case the estimated value of the random factor parameter 'respondents' ID [48]. Values less than 0.50 indicate poor reliability, between 0.50 and 0.75 moderate reliability, between 0.75 and 0.90 good reliability and greater than 0.90 excellent reliability. Because we expect that the Trust game decisions show substantial variability over time, values less than 0.50 would support our hypothesis.

To test the hypothesis that social trust questionnaire responses measured on the first and last session of the experiment show substantial stability over time, a correlation analysis will be used. We will use the standard statistical significance threshold of $p < 0.05$.

To test the main hypothesis that the average behaviour in the Trust game on the first session is not or poorly correlated with social trust questionnaire responses while the average behaviour in the Trust game over all sessions is significantly and more strongly correlated with social trust questionnaire responses, a mixed effects model will be performed with the R package *nlme* and fitted with a Maximum-Likelihood method. The willingness-to-play averaged on the first session and over all sessions is the dependent variable, labelled AWP (Average Willingness to Play). The single responses that give rise to AWP are measured via a nine-point Likert scale, and in principle could be classified as ordered categorical. However, AWP represents the average of these responses (16 on session 1; 160 over all sessions) and therefore includes a number of ranks that is large enough to consider it as approximately continuous. Note in addition that responses measured on Likert scales with five or more points share many properties with continuous data and can be treated as such [49–52]. This leads us to test a model where AWP will be treated as a (approximately) continuous dependent variable, the questionnaire responses (Qavg) as a (approximately) continuous predictor, and the factor Session as a binary variable (0 = S1; 1 = Sall). Age will be included as a covariate, because it has been shown to positively correlate with generalized trust levels and Trust game decisions [53,54]. All these variables will be standardized (i.e. *z*-scored). In addition, the respondents' IDs will be included as a grouping factor with a random intercept. Finally, an interaction term between Qavg*Session will be included to test whether the correlation of Qavg with AWP observed at D1 differs from the correlation of Qavg and AWP observed at Dall. We will use the standard statistical significance threshold of $p < 0.05$. A correlation slope difference of 0.20 will be taken as the effect size of reference to consider the alternative hypothesis as validated. Two simple linear regression models will be used as post-hoc analyses to estimate the slopes characterizing the association of Qavg and AWP at S1, and at Sall, respectively.

## 2.7. Preregistration: power analysis and sample-size estimation

To determine sample size, a power analysis was performed using Monte Carlo simulations with the R package *simr* [55]. Monte Carlo simulations are particularly useful to calculate power and sample-size requirements when analytical solutions cannot be derived because of the complexity of the target statistical model. Using the parameters of the distributions of the Trust game decisions and the

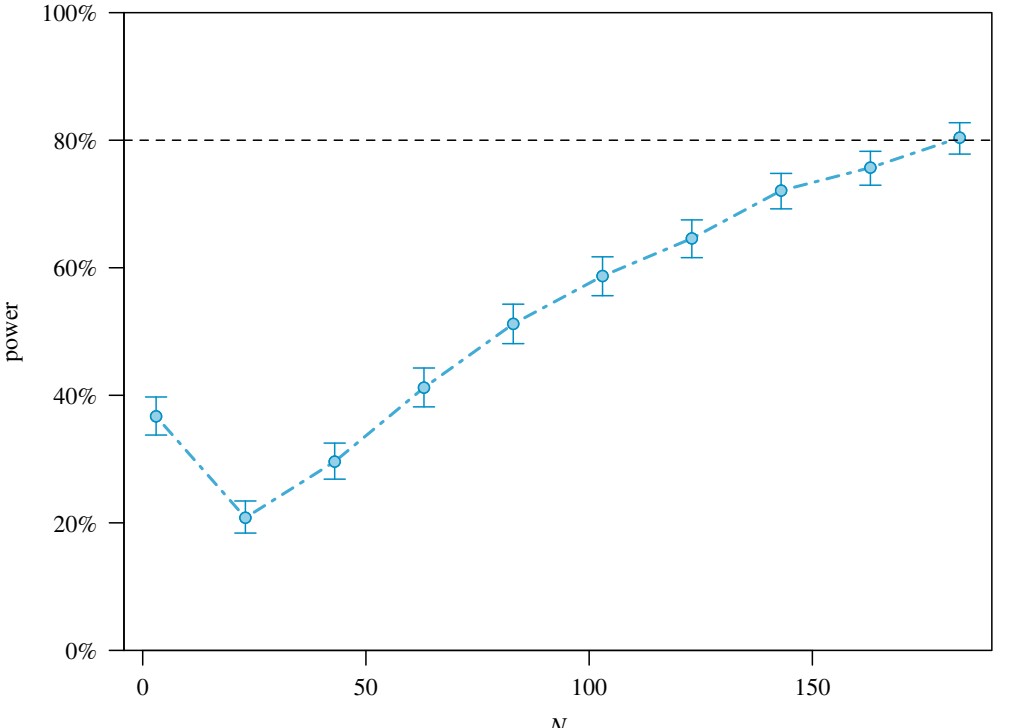

**Figure 1.** Power curve to detect an effect size of 0.20. This figure shows that 183 respondents are needed to reach a power of 80%.

questionnaire responses of Mell *et al*. [34], we conducted an *a priori* power analysis of the effect of the interaction Qavg*Session on the AWP variable, mentioned in the description of our main analytic model above, and which tests the difference between the correlation of Qavg with AWP observed at S1 and the correlation of Qavg and AWP observed at Sall. A slope difference of 0.20 (which is considered moderate according to standard guidelines, see [56]) was chosen as the minimum effect size detectable with 80% power. This moderate effect size was chosen because we are only interested in an effect where it is useful for other researchers to use our methodology (i.e. repeating economic games over time) in further research. If we only find a weak effect, then the added value of using repeated economic games over time might not weigh up against the additional cost and time to perform such a methodology. The Monte Carlo simulations power analysis indicated that a minimum of 183 respondents are required to detect a slope difference effect of 0.20 with 80% power (figure 1).

In addition, an independent study [57] found that for a longitudinal study (i.e. 12 months) performed on Prolific the attrition rate was around 25%. Because our study is significantly shorter, we expect a lower attrition rate of 10%. Furthermore, in a study with a similar methodology [34], 25% of the respondents were excluded from the analysis for not correctly answering the questions in the last of the two comprehension check sections. To compensate for the expected attrition and exclusion of respondents (35% in total), we will recruit 275 respondents.

## 2.8. Preregistration: exploratory analyses

In addition to the pre-registered analyses detailed above, additional exploratory analyses will be performed. First, concerning real-world behaviour related to social trust (i.e. data sharing and actual helping) and the Trust game decisions, we will use mixed effects models in a similar manner to that described above. The only difference is that real-world behaviour will be included as a binary variable (0 = not sharing or helping; 1 = sharing or helping) and the average of the questionnaire responses (Qavg) will be removed. To test the relationship between real-world behaviour and the questionnaire responses we will use a one-way mixed effects model, where the average of the questionnaire responses (Qavg) will be the dependent variable and real-world behaviour is an explanatory binary variable (0 = not sharing or helping; 1 = sharing or helping).

Second, we will also perform analyses exploring the dynamic structure of the behaviour of respondents playing the repeated Trust games. At this stage, we do not have specific hypotheses about the precise pattern of variation that we can expect from the respondents' behaviour over time, nor about whether and how the structure of the pattern itself relates to the responses in the social trust questionnaire. For example, is there an increase or decrease in AWP over time? Furthermore, how many sessions of playing Trust games are needed before a significant correlation with the questionnaires emerges, measured using mixed effects models. Finally, latent class analysis (LCA) could be employed to identify subgroups of respondents that might respond differently to the longitudinal design based on multiple characteristics (e.g. average level of social trust measured by the questionnaire). LCA [58–60] is a mixture model that posits that there is an underlying unobserved categorical variable that divides a population into mutually exclusive and exhaustive latent classes. Class membership of individuals is unknown but can be inferred from a set of measured items. Thus, LCA might help us identify subgroups of respondents each displaying a specific pattern of behavioural variability over time.

## 2.9. Tested participant sample

Of the 275 participants included in the study on Prolific.co, 155 completed the 10 sessions of the longitudinal study (84 women, 70 men; mean age = 34.65, s.d. = 8.23). The mean bonus payment across the 10 sessions matched the expected range (actual mean bonus payment: 5.64£, predicted mean bonus payment: 5.75£). One participant was excluded from the analyses following the pre-registered exclusion procedure. The analyses were thus conducted on 154 participants (1 non-binary person, 83 women, 70 men; mean age = 34.66, s.d. = 8.25). This sample size is lower than the targeted sample size of 183 respondents. However, based on our power calculations, this sample size is sufficient to detect a moderate slope difference in the association of participants' behaviour in the Trust game and their responses to the social trust questionnaire between data collected on Session 1 and the data averaged across the 10 sessions with a power of 73.80%. Importantly, the final sample did not significantly differ from the sample of participants who did not complete the full experiment in terms of social trust (both measured by the social trust questions, $t_{272} = 1.39$, $p = 0.164$, and the trust game in Session 1: questionnaire: $t_{272} = -1.32$, $p = 0.187$).

# 3. Results

## 3.1. Positive control checks

The average willingness-to-play on Session 1 was within the expected range calculated based on Mell *et al*. [34] (expected range: [4.62–7.44], average willingness-to play-on Session 1, Mean = 5.75, s.d. = 1.35), which validates our positive control.

## 3.2. Data quality check and data cleaning

The average willingness-to-play to the Trust Games did not deviate significantly from a normal distribution at the 0.05 significance level (Session 1: $W = 0.98$, $p = 0.061$; all sessions combined: $W = 0.98$, $p = 0.079$).

However, the distribution of responses to the social trust questionnaire deviated significantly from a normal distribution (Session 1: $W = 0.98$, $p = 0.013$; Session 10: $W = 0.98$, $p$-value = 0.020; see electronic supplementary materials, S4 for P-P plots). Nevertheless, inter-item reliability (i.e. Cronbach's alpha) indicated high internal consistency of our three social trust questions (Session 1: alpha = 0.80; Session 10: alpha = 0.83).

## 3.3. Pre-registered analyses

In line with our hypothesis, the ICC measure confirmed our hypothesis of low reliability of the decisions in the Trust Games over time (ICC = 0.11). On the other hand, the correlation between the social trust measured by questionnaires in Session 1 and 10 was high (Spearman's rho = 0.62, $S = 140852$, $p < 0.001$) confirming a substantial stability of these measures.

Contrary to our prediction, however, the regression on the willingness-to-play in the Trust Game during Session 1, and then averaged over all 10 sessions, revealed a significant positive effect of self-reported social trust measured by questionnaires (averaged on Session 1 and Session 10; $b = 0.19 \pm 0.09$ s.e.m., $t_{151} = 2.12$,

**Table 1.** Association between willingness-to-play and social trust measured by questionnaires.

| | session 1 | all sessions combined | session 1 and all sessions combined |
|---|---|---|---|
| intercept | $b = 0.04 \pm 0.09$ s.e.m. | $b = -0.04 \pm 0.07$ s.e.m. | $b = 0.04 \pm 0.08$ s.e.m. |
| | $t_{151} = 0.52$ | $t_{151} = -0.63$ | $t_{152} = 0.57$ |
| | $p > 0.250$ | $p > 0.250$ | $p > 0.250$ |
| social trust questionnaire | $b = 0.19 \pm 0.09$ s.e.m. | $b = 0.22 \pm 0.07$ s.e.m. | $b = .20 \pm 0.08$ s.e.m. |
| | $t_{151} = 2.12$ | $t_{151} = 2.94$ | $t_{151} = 2.38$ |
| | $p = 0.036$ | $p = 0.004$ | $p = 0.019$ |
| session (Session 1 versus all sessions combined) | | | $b = -0.09 \pm 0.07$ s.e.m. |
| | | | $t_{152} = -1.24$ |
| | | | $p = 0.216$ |
| age | $b = .13 \pm 0.09$ s.e.m. | $b = 0.11 \pm 0.07$ s.e.m. | $b = 0.11 \pm 0.07$ s.e.m. |
| | $t_{151} = 1.42$ | $t_{151} = 1.43$ | $t_{151} = 1.59$ |
| | $p = 0.158$ | $p = 0.155$ | $p = 0.112$ |
| social trust questionnaires : session | | | $b = 0.02 \pm 0.07$ s.e.m. |
| | | | $t_{152} = 0.26$ |
| | | | $p > 0.250$ |

$p = 0.036$). However, this association was not stronger when considering average willingness-to-play over the 10 sessions (interaction effect: $b = 0.02 \pm 0.07$ s.e.m., $t_{152} = 0.26$, $p > 0.250$; table 1). Specifically, the analysis of willingness-to-play on Session 1 and across all sessions revealed a positive slope difference of only $0.02 \pm 0.07$ s.e.m. (table 1). To summarize, aggregating participants' behaviour in the trust game over a 20-day period did not significantly increase the slope of the association between behavioural and psychometric measures of social trust.

## 3.4. Pre-registered exploratory analyses

### 3.4.1. Association between experimentally measured behaviours and real-world behaviours

In line with our analysis plan, we tested whether participants' average behaviour in the trust game over a 20-day period increased the association between experimentally measured social trust and real-world behaviour, i.e. willingness to help or share personal data with researchers. No significant association between experimentally measured behaviour and real-world behaviour was found (both $p$-s $> 0.250$). The association between real-world behaviour and social trust measured with questionnaires was not found neither ($b = 0.77 \pm 0.68$ s.e.m., $t_{151} = 1.13$, $p > 0.250$).

### 3.4.2. Heterogeneous trajectories of willingness-to-play

Latent Class Mixed Models were then conducted using the 'hlme' function of the *lcmm* R package [61] in order to define subpopulations of participants ($N = 154$) who displayed various trajectories of willingness-to-play trajectory over the entire experiment [62]. The metric we used to analyse individuals' willingness-to-play trajectories was the number of days participants spent between the 1st and the 10th economic game session, and was coded as follows: day 1, day 3, day 5, day 7, day 9, day 11, day 13, day 15, day 17, day 19). Importantly, the time interval between two game sessions was held constant along the entire experiment and across subjects. Note in addition that no missing value was present in this longitudinal dataset. The average willingness-to play-collected on each day departed from normality, so the whole vector of willingness-to-play data was z-scored. The procedure we used replicates the procedure described in detail by Edjolo *et al.* [63], and follows the Guidelines for Reporting on Latent Trajectory Studies (GRoLTS; [64]). The sample was divided in a finite number $G$ of subpopulations defining latent classes with a discrete latent variable $c_i$ ($c_i = g$ if $i$ belongs to the $g$ class). For each participant, a probability of belonging to a unique latent class was computed via a

multinomial logistic model and, for each class, the dynamic of the dependent variable was described with a linear mixed model. After the estimation process, posterior latent class membership probabilities were computed and provided a classification of each participant according to the profile that best matched their data. The models were estimated with the maximum-likelihood method, and for a fixed number of six latent classes [65]. Since we have no *a priori* reason to expect the trajectories to suddenly decelerate or accelerate as the end of the experiment approaches, we considered linear instead of cubic or quadratic shapes of trajectories over time to account for the general increase, decrease or stability of willingness-to-play. Initial starting values were determined as the output of a grid search process whereby a given model was iteratively replicated on a parameter space comprising 100 random vectors. Two correlated individual random effects (on the intercept and the slope) captured the inter-participant variability in all models. Given the small size of our sample, the optimal number of classes was identified by selecting the model with the lowest sample-size adjusted BIC (SABIC) [66], in conjunction with the size of the classes and the discriminatory performances (the posterior probabilities as indication of participants' class membership). Although the two-class model reported the lowest SABIC (SABIC = 3 417.68), it was made of two very unbalanced classes (class 1 = 99% ; class 2 = 1%) and, as such, did not add information respective to the baseline 1-class solution (SABIC = 3 419.7). We therefore retained the 4-class model (SABIC = 3418.05), which is nearly equivalent in terms of fitting, but which identifies two classes with 50 participants or more (i.e. 62% and 32% of the total sample, table 2).

In this model, the third class included seven participants (5% of the total sample), and the fourth class captured one participant. This single profile was captured by each of the *G*-class models, suggesting that it is an outlier profile. The 4-class model also yielded a high classification precision, with mean posterior probabilities of 0.905 for class 1, 0.910 for class 2, 0.878 for class 3 and 0.998 for class 4.

The class analysis captured four patterns of willingness-to-play trajectories. The first class, which involved the largest number of participants ($N = 96$), was characterized by a slow decrease in willingness-to-play of 0.31 s.d. unit from the sample mean (intercept = −0.093) as the days went by, and was labelled 'willingness-to-play slow decrease'. The second class ($N = 50$) was characterized by a moderate increase in willingness-to-play of 0.786 s.d. unit from the sample mean (intercept = 0.312). We labelled it 'willingness-to-play increase'. The third class showed a steep willingness-to-play decrease of −1.591 s.d. unit from the sample mean (intercept = −0.437), but only included seven participants. It was labelled 'willingness-to-play steep decrease'. Finally, the fourth class captured the trajectory of a single participant who featured an extremely low and stationary willingness-to-play. We labelled it the 'willingness-to-play null' profile.

### 3.4.3. Predictors of participants' class membership

We then investigated whether our psychometric (Social trust questionnaire average score) and real-world (Extra help and Data sharing) proxies of social trust predicted the class to which participants were more likely to belong. We also studied whether age, gender, and current socio-economic status predicted class membership. We used the *nnet*, *broom*, *scales* and *car* R packages [67–70] to perform multinomial logistic regressions weighted by the logit-transformed posterior probabilities of class membership [71]. Because only one participant identified as non-binary, we limited our analyses to participants who identified as either a woman or man to avoid misinterpretation of differences associated with this category. Similarly, the single participant in the 'willingness-to-play null' class was excluded from the analyses. The analyses were therefore run on a sample of $N = 152$ participants. Each class predictor was considered in a univariate analysis first, and then included in a backward stepwise multivariate multinomial logistic model when a significant effect was found. The class of reference was 'willingness-to-play slow decrease' in both the univariate and multivariate multinomial logistic models, as it includes the largest number of participants.

Compared to a baseline model including only an intercept, adding either the questionnaire scores (LLR test: $\chi^2 = 27.24$; d.f. = 2; $p < 0.001$), age (LLR test: $\chi^2 = 10.48$; d.f. = 2; $p = 0.005$) or gender (LLR test: $\chi^2 = 8.23$; d.f. = 2; $p = 0.016$), increased the goodness of fit of the model (see electronic supplementary material S5, table S7). Neither real-world social trust behaviours (Extra help, Data sharing) nor current SES increased the models' fit (see electronic supplementary material S5, table S7).

Among the three predictors selected at the univariate step (questionnaire scores, age and gender), only questionnaire scores and gender remained significantly associated with willingness-to-play once all three predictors were entered in a multivariate model (see electronic supplementary material, S5, tables S8 and S9). On average, as the mean score on the social trust questionnaire increased, a participant was more

**Table 2.** Results of the Latent Class Mixed Models. The solution that provides the best trade-off between fitting values, classes' distribution, size and specificity is a 4-class model.

| model | G | loglik | conv | npm | SABIC | %class1 | %class2 | %class3 | %class4 | %class5 | %class6 |
|---|---|---|---|---|---|---|---|---|---|---|---|
| 1-class | 1 | −1704.23 | 1 | 6 | 3419.68 | 100.00 | | | | | |
| 2-class | 2 | −1700.13 | 1 | 9 | 3417.11 | 99.35 | 0.65 | | | | |
| 3-class | 3 | −1699.11 | 1 | 12 | 3420.69 | 98.70 | 0.65 | 0.65 | | | |
| **4-class** | **4** | **−1694.99** | **1** | **15** | **3418.05** | **62.34** | **0.65** | **32.47** | **4.55** | | |
| 5-class | 5 | −1693.70 | 1 | 18 | 3421.09 | 62.34 | 31.82 | 0.65 | 4.55 | 0.65 | |
| 6-class | 6 | −1693.74 | 1 | 21 | 3426.78 | 43.51 | 20.78 | 3.90 | 29.22 | 0.65 | 1.95 |

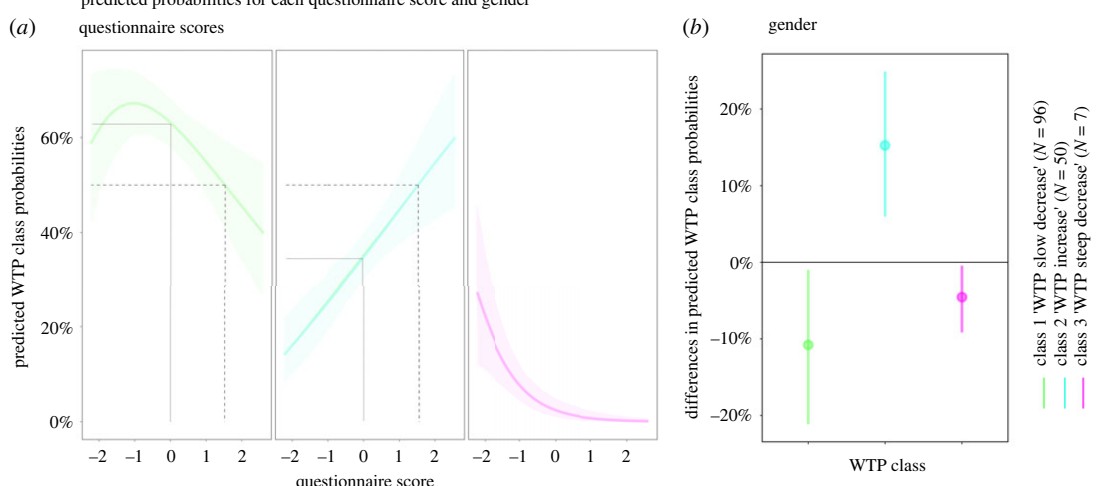

**Figure 2.** Effects of predictors of participants' willingness-to-play class membership as obtained from the best-fitting multinomial logistic regression model. (*a*) Effect of standardized social trust questionnaire scores: slopes indicate the probability (and the 95% CIs) of being classify a given willingness to play trajectory profile for every possible standardized scores (in standard deviation units from the group mean) on the social trust questionnaire (averaged on the first and last session). The dashed lines indicate the questionnaire scores at which the predicted probabilities of class membership are equal to 0.5. The full lines indicate predicted probabilities for a questionnaire of 0 (unstandardized group mean score = 4.76). (*b*) Effect of gender: whiskers represent the differences in the predicted probability (and 95% CIs) of being classifies as in given willingness-to-play trajectory profile for female participants, when compared to a male participant.

likely to be classified in the 'willingness-to-play increase' than in the 'willingness-to-play slow decrease' trajectory profile (RRR = 0.40; s.e. = 0.12; z = 3.39; OR = 1.50; CI lower = 1.19; CI upper = 1.90; $p < 0.001$). Conversely, as the mean score on the social trust questionnaire decreased, a participant was more likely to be classified in the 'willingness-to-play steep decrease' than in the 'willingness-to-play slow decrease' trajectory profile (RRR = −1.18; s.e. = 0.33; z = −3.55; OR = 0.31; CI lower = 0.16; CI upper = 0.59; $p < 0.001$). In addition, the model showed that female participants were more likely to be classified in the 'willingness-to-play increase' profile (RRR = 0.67; s.e. = 0.24; z = 2.82; OR = 1.95; CI lower = 1.23; CI upper = 3.10; $p = 0.004$) than in the 'willingness-to-play slow decrease' profile.

In order to investigate more precisely how the probability of being assigned into a specific class of willingness-to-play trajectory varied for each questionnaire score, we used the *MNLpred* R package [72]. Inspection of predicted probabilities revealed that the likelihood of being classified in the 'willingness-to-play increase' profile linearly increased with scores on the social trust questionnaire, starting at 0.14 for the lowest score and culminating at 0.60 for the highest score (figure 2*a*). Conversely, the first half of the probability curve of the 'willingness-to-play slow decrease' profile followed a quadratic pattern for questionnaires scores that were inferior to the group mean. The curve started at a probability of 0.60 for the lowest score and ended at a probability of 0.62 for the group mean questionnaire score of 0, with an in-between peak at 0.69. Instead, the second half of the curve—from the group mean questionnaire score to the highest score—showed a linear decrease ending at a probability of 0.39. Concerning the 'willingness-to-play steep decrease' profile, predicted probabilities dropped exponentially as long as questionnaire scores increased, starting at a probability of 0.27 and ending at a 0 probability. An interesting phenomenon is that being classified as 'willingness-to-play increase' or 'willingness-to-play slow decrease' was equiprobable for a standardized questionnaire score of 1.52 (figure 2*a*, dashed lines). This observation suggests that higher scores on the social trust questionnaire are more prone to measurement noise, making them less efficient to grasp the psychological construct they are made for (see the Conclusion section).

Inspection of predicted probabilities calculated for each gender (figure 2*b*) revealed that the probability of being classified as 'willingness-to-play increase' was 15% higher for a female participant than for a male participant (0.42 versus 0.26). Compared to male participants, the probability for a female participant to be classified as 'willingness-to-play slow decrease' and 'willingness-to-play steep decrease' was 11% (0.67 versus 0.66) and 5% lower (0.07 versus 0.02).

## 3.5. Unregistered analyses

Descriptive statistical analyses revealed that more than 98% of our participant sample agreed to either share personal data with other researchers or to provide comments on our study. This small variability in real-world cooperative behaviour may explain the lack of a significant association with self-reported social trust using questionnaires. As only 70% of the participants sampled decided to provide comments on the study while 97% agreed to share their data, we tested this hypothesis by conducting the planned analyses on the former measure of real-world social trust. However, these new analyses revealed no significant association with the willingness-to-play (Session 1: $b = 0.01 \pm 0.20$ s.e.m., $t_{151} = 0.03$, $p > 0.250$; averaged across the sessions: $b = 0.13 \pm 0.16$ s.e.m., $t_{151} = 0.77$, $p > 0.250$), nor with self-reported trust measured using questionnaires ($b = 0.09 \pm 0.18$ s.e.m., $t_{151} = 0.54$, $p > 0.250$).

# 5. Conclusion

The goal of this study was to assess whether erasing day-to-day variability would increase the reliability of economic games to measure cooperative tendencies. Our study revealed a significant association between self-reported social trust and participants' behaviour in the trust game measured on Session 1 and averaged across 10 sessions spanning 20 days. As predicted, the measure of trust behaviour was not stable over time. However, our analyses failed to reveal a stronger association between willingness-to-play in the trust games averaged across all 10 sessions and willingness-to-play measured during the first session of the study.

It is worth noting that one of our pre-registered hypotheses was responses to the questionnaires and willingness-to-play during the first session would not be correlated. Instead, a correlation was found in Session 1, which may explain why averaging willingness-to-play across all sessions did not significantly increase the association between willingness-to-play and self-reported social trust. This result may be due to the use of a repeated Trust Game, instead of the classical one-shot Trust Game. Indeed, repeated trust decisions towards partners varying in their trustworthiness and required stakes may allow for measuring more stable cooperative traits than one-shot games.

In addition, it is important to note that the present design only tested participants during a relatively short time period (three weeks). As a consequence, our study only allowed us to assess potential statistical effects of reducing the noise due to day-to-day variations and not longer-term variations. Therefore, further experiments should be conducted in order to assess whether the association between behaviour in the trust games and self-reported social trust increases over longer timescales.

In spite of these negative results, we showed through a set of supplementary latent class mixed models and multivariate multinomial regressions that the self-reported measure of social trust was associated with individual trajectories in the trust game measured over the three-week period. In particular, two meaningful classes of trajectories were identified, one ascending (willingness-to-play increase) and one descending (willingness-to-play slow decrease), within which 94.8% of the participants were distributed. Overall, subjects with higher scores on the social trust questionnaire were more likely to be assigned to the ascending class—i.e. where willingness-to-play increased over the 10 sessions of the experiment—whereas those with lower scores on the social trust questionnaire were more likely to be assigned to the descending class—i.e. where willingness-to-play decreased over the 10 sessions of the experiment. Interestingly, the probabilities of assignment to these two classes were equal for participants who scored the highest on the social trust questionnaire (1.52 in standardized units), whereas they were clearly distinct for those who scored the lower (figure 2a). One possible interpretation of this phenomenon is that the repeated trust game is better suited to capture the level of long-term engagement in the task. This is in line with empirical data showing a positive relationship between social trust and personality traits like conscientiousness, openness to new experience and agreeableness [73]. Despite obvious advantages, the social trust questionnaire would instead be more prone to social desirability bias, i.e. when participants try to portray themselves in a more socially favourable light.

These supplementary analyses further revealed that the behavioural dynamic at stake in the repeated social trust game differed between male and female participants. Men were indeed more likely than women to be classified in the two descending profiles (willingness-to-play slow decrease and willingness-to-play steep decrease). We also observed that the distribution of willingness-to-play trajectory profiles slightly diverged between male and female participants (figure 3b). Indeed, the sub-sample of male participants featured a clear dominance of the descending patterns, whereas the 'willingness-to-play slow decrease' and 'willingness-to-play increase' profiles were more homogeneously distributed within the

sub-sample of female participants. Such a gender effect in human trust and cooperation research is well-documented [74] and may be the product of evolutionary pressures imposed on women which might have favoured the selection of traits that facilitate social support networks building [75], such as trust and cooperation.

Our data did not reveal any association between experimental social trust measures and real-world behaviour. However, this null effect may be driven by a ceiling effect in the real-life measure included in the study.

Overall, our study questions the use of existing tools to measure social trust, rather than their validity. One key point is to understand the extent to which self-reported measures and experimental measures can be related to one another and to real-life behaviour but also to identify the strengths and weaknesses of these different tools. As it has been previously argued, experimental tasks have often been designed to maximize individual sensitivity to different experimental conditions. In that perspective, they are the tool best suited to understanding which contextual factors affect social trust behaviour but not necessarily to predicting actual individual behaviour in real-life [24]. In particular, the modified trust game used in this experiment allows for the investigation of within-game individual variations, which is not the case for most of the games in behavioural economics. As a result, our game may have a lower ability to identify inter-individual variations. Therefore, even if it appears that social trust behaviours measured using different tools are related, further work is needed to precisely identify which tool is best suited to test specific hypotheses.

Ethics. The experimental protocol is approved by the local Ethical Committee (Conseil d'évaluation éthique pour les recherches en santé - CERES no. 201659). Furthermore, an informed consent form will be signed by respondents.
Data accessibility. The Stage 1 Registered Report (written prior to any data collection) can be found on the following link: https://osf.io/fa697. The data and analyses scripts are openly accessible on the Open Science Framework: https://osf.io/ht863.

Supplementary material is available online [76].
Authors' contributions. L.S.: conceptualization, data curation, formal analysis, investigation, methodology, project administration, visualization, writing—original draft, writing—review and editing; N.L.: conceptualization, formal analysis, investigation, methodology, writing—original draft, writing—review and editing; P.J.: conceptualization, data curation, formal analysis, investigation, methodology, project administration, supervision, validation, visualization, writing—original draft, writing—review and editing; C.C.: conceptualization, formal analysis, funding acquisition, investigation, methodology, project administration, resources, software, supervision, validation, writing—original draft, writing—review and editing.

All authors gave final approval for publication and agreed to be held accountable for the work performed therein.
Conflict of interest declaration. We declare we have no competing interests.
Funding. This study was supported by the EUR FrontCog grant no. ANR-17-EURE-0017 and the grant ANR-21-CE28-0009.

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
