## [Peer Review File · Royal Society Open Science]

Review History

RSOS-210213.R0 (Original submission)

Review form: Reviewer 1

Do you have any ethical concerns with this paper?

Yes

Recommendation?

Major revision

Comments to the Author(s)

See PDF (Appendix A).

Review form: Reviewer 2

Do you have any ethical concerns with this paper?

No

Recommendation?

Major revision

Comments to the Author(s)

This is an interesting and well worked-out study registration: The research questions are clear and topical, the proposed design in principle make sense (but see below), and the analysis plan is mostly solid. In what follows I will list a series of comments and questions that hopefully will be useful to further strengthen this study. Most of these are relatively minor comments, but one comment is more fundamental and requires particular attention, which is why I raise this issue at the outset.

Specifically, the entire study is about measuring social trust – but the authors essentially fool their participants by telling them that they will play each game "on the fly" with another real participant (who is actually even supposed to be substituted with another real participant from trial to trial!). In reality, however, it is only the computer playing with the actual participants. The "data-sharing test" in the end is another deception, as participants who agree to share their data will be told that "they were not selected" (and in reality the data of nobody will ever be shared). I understand that there are some intricacies when studying social trust, but with the current implementation I see the real danger that participants will not buy the story that they are indeed playing with real (and different) partners on the fly (e.g., they will never have to wait for the response of the other player, making it quite easy to detect that some deception is going on). Thus, assuming that participants realize that they were tricked and actually play with a machine, why should they themselves trust the experimenters / this machine? And thus, can one indeed expect that participants will ultimately behave consistently / realistically? Importantly, there exist various internet forums in which participants quickly exchange insights gained in such online studies; hence there may be backfiring effects for this experiment specifically, but also for similar future studies more generally (see Hertwig & Ortmann, 2001). With current technologies it would be possible, in principle, to set up a study on social trust without any deception, including on platforms such as mTurk or Prolific (e.g., for a truly interactive online game see Frey, 2020). All in all, this issue definitely requires some further deliberation and potentially the consideration of alternative empirical approaches.

* Scientific validity of the research question(s)

This study addresses a topical and important research question, and I only have three (relatively minor) comments in this regard:

1) The authors suggest three "advantages" of economic games on p. 3. Yet, as will become clear just a couple of paragraphs below, at least some of these "advantages" may rather be disadvantages. For instance, to the extent that games indeed capture "actual behavior" ("advantage 1") and assuming that this behavior is highly variable, this is evidently a disadvantage provided that one aims to study a trait. Moreover, exactly due to the lack of reliability as introduced shortly afterwards, games may in fact not be very well suited to "precisely investigate inter-individual differences" ("advantage 3"). These advantages should thus either be introduced more thoroughly and convincingly, or potentially be omitted altogether.

2) Generally the authors argue that games may be more susceptible to state influences whereas self-reports may rather capture a stable trait. Although recent evidence indeed suggests a series

of cognitive explanations for the latter hypothesis (e.g., Steiner et al., 2021), some questionnaires may actually be designed to specifically capture states, too (e.g., the state-trait anxiety inventory was developed to assess situational and general dispositions; Spielberger, 2010).

3) Relatedly, the authors appear to presuppose that social trust is a stable personality disposition. But is trust not highly variable across different contexts, as a function of different personal interactions, etc.? In other words, is the expected variability in trust (as measured by the repeated games; hypothesis 1) simply considered random noise, or is this the result of a systematic mechanism? Overall a more detailed elaboration on how the authors conceptualize social trust would be helpful.

* Logic, rationale, and plausibility of the proposed hypotheses

The hypotheses are plausible and clearly specified.

* Soundness and feasibility of the methodology and analysis pipeline (including statistical power analysis where applicable)

Generally the empirical approach is sound, but I have a series of comments in this regard.

1) The exclusion criteria concerning "obvious random responses" should be specified more clearly.

2) Participant recruitment will be stopped "once the target sample is reached"; this requires that the specified quality-control / exclusion criteria will have to be monitored on the fly (otherwise the final sample will be small than the aspired sample size). Is this the case?

3) Is there any concern for anchoring effects, given that participants fill in the questionnaire in the first session? What about splitting the sample into two groups (a: questionnaires pre & post repeated assessments; b: questionnaire only post repeated assessments)?

4) (How) will it be ensured that participants play the games only every second day?

5) I was confused by the statement "In the 8 intermediate days, respondents only play the Trust game 16 times" (p. 8). Why "only"? They also play 16 rounds in the initial session, or do they play more (if so, please clarify)?

6) The authors plan to use ICCs to quantify the reliability of the economic games. Instead of computing 16 separate ICCs and then averaging them, I wondered whether one could implement a single model that treats the 16 games (or even the 4 x 4 predictors) as fixed effects, to thus directly extract one overall indicator of intraindividual variability?

7) Although the analysis plan is specified quite well, but the authors should spell out explicitly under which conditions (thresholds for statistical tests, etc.) they consider their hypotheses confirmed.

* Whether the clarity and degree of methodological detail would be sufficient to replicate exactly the proposed experimental procedures and analysis pipeline

Yes, but see points above.

* Whether the authors provide a sufficiently clear and detailed description of the methods to prevent undisclosed flexibility in the experimental procedures or analysis pipeline

Yes, but see points above.

* Whether the authors have considered sufficient outcome-neutral conditions (e.g. positive controls) for ensuring that the results obtained are able to test the stated hypotheses

Yes, but there is some ambiguity as to how "close" is "close enough" compared to the results of Mell et al. (p. 11); please specify.

References:

Frey, R. (2020). Decisions from experience: Competitive search and choice in kind and wicked environments. *Judgment and Decision Making*, 15(2), 282–303.

Hertwig, R., & Ortmann, A. (2001). Experimental practices in economics: A methodological challenge for psychologists? *Behavioral and Brain Sciences*, 24(03), 383–403.

Spielberger, C. D. (2010). State-Trait anxiety inventory. *The Corsini encyclopedia of psychology*.

Steiner, M., D., Seitz, F., & Frey, R. (in press). Through the window of my mind: Mapping information integration and the cognitive representations underlying self-reported risk preference. *Decision*. <https://doi.org/10.31234/osf.io/sa834>

Decision letter (RSOS-210213.R0)

Dear Dr Lettinga,

The Editors assigned to your stage one Registered Report ("Variability in repeated economic games; comparing Trust game decisions to other social trust measures") have now received comments from reviewers. We would like you to revise your paper in accordance with the referee and editors suggestions which can be found below (not including confidential reports to the Editor). Please note this decision does not guarantee eventual acceptance.

Please submit a copy of your revised paper within three weeks (i.e. by the 30-Mar-2021). If we do not hear from you within this time then it will be assumed that the paper has been withdrawn. If deemed necessary by the Editors, your manuscript will be sent back to one or more of the original reviewers for assessment. If the original reviewers are not available we may invite new reviewers.

When submitting your revised manuscript, you must respond to the comments made by the referees and upload a file "Response to Referees" in "Section 2 - File Upload". Please use this to document how you have responded to the comments, and the adjustments you have made. In

order to expedite the processing of the revised manuscript, please be as specific as possible in your response.

Kind regards,
Professor Chris Chambers
Royal Society Open Science
openscience@royalsociety.org

on behalf of Professor Chris Chambers (Registered Reports Editor, Royal Society Open Science)
openscience@royalsociety.org

Associate Editor Comments to Author (Professor Chris Chambers):

Comments to the Author:

Two expert reviewers have now assessed the Stage 1 manuscript and have provided very constructive and detailed reviews. Both reviewers find merit in the proposal while also highlighting a wide range of issues that will need to be addressed to progress the manuscript toward in-principle acceptance. Headline issues include the theoretical framing of the research (and level of theoretical depth), concerns with the deception manipulation (both ethical and also scientific in terms of its effectiveness), concerns with self-report measures, the validity of the proposed analysis pipeline, and the clear linking of the predictions with the analyses that will confirm or disconfirm them. The reviews also contain many detailed comments concerning the proposed methods and the need for greater clarification and justification of design decisions.

Reviews this critical would likely lead to rejection of a completed study, but since the RR process gives authors to opportunity to address serious concerns before research is undertaken, I would like to offer the authors the opportunity to submit a Major Revision. Any revised manuscript will be returned to the reviewers for reassessment.

Comments to Author:

Reviewer: 1

Comments to the Author(s)

See PDF

Reviewer: 2

Comments to the Author(s)

This is an interesting and well worked-out study registration: The research questions are clear and topical, the proposed design in principle make sense (but see below), and the analysis plan is mostly solid. In what follows I will list a series of comments and questions that hopefully will be useful to further strengthen this study. Most of these are relatively minor comments, but one comment is more fundamental and requires particular attention, which is why I raise this issue at the outset.

Specifically, the entire study is about measuring social trust – but the authors essentially fool their participants by telling them that they will play each game "on the fly" with another real participant (who is actually even supposed to be substituted with another real participant from trial to trial!). In reality, however, it is only the computer playing with the actual participants. The "data-sharing test" in the end is another deception, as participants who agree to share their data will be told that "they were not selected" (and in reality the data of nobody will ever be shared). I understand that there are some intricacies when studying social trust, but with the current

implementation I see the real danger that participants will not buy the story that they are indeed playing with real (and different) partners on the fly (e.g., they will never have to wait for the response of the other player, making it quite easy to detect that some deception is going on). Thus, assuming that participants realize that they were tricked and actually play with a machine, why should they themselves trust the experimenters / this machine? And thus, can one indeed expect that participants will ultimately behave consistently / realistically? Importantly, there exist various internet forums in which participants quickly exchange insights gained in such online studies; hence there may be backfiring effects for this experiment specifically, but also for similar future studies more generally (see Hertwig & Ortmann, 2001). With current technologies it would be possible, in principle, to set up a study on social trust without any deception, including on platforms such as mTurk or Prolific (e.g., for a truly interactive online game see Frey, 2020). All in all, this issue definitely requires some further deliberation and potentially the consideration of alternative empirical approaches.

* Scientific validity of the research question(s)

This study addresses a topical and important research question, and I only have three (relatively minor) comments in this regard:

1) The authors suggest three "advantages" of economic games on p. 3. Yet, as will become clear just a couple of paragraphs below, at least some of these "advantages" may rather be disadvantages. For instance, to the extent that games indeed capture "actual behavior" ("advantage 1") and assuming that this behavior is highly variable, this is evidently a disadvantage provided that one aims to study a trait. Moreover, exactly due to the lack of reliability as introduced shortly afterwards, games may in fact not be very well suited to "precisely investigate inter-individual differences" ("advantage 3"). These advantages should thus either be introduced more thoroughly and convincingly, or potentially be omitted altogether.

2) Generally the authors argue that games may be more susceptible to state influences whereas self-reports may rather capture a stable trait. Although recent evidence indeed suggests a series of cognitive explanations for the latter hypothesis (e.g., Steiner et al., 2021), some questionnaires may actually be designed to specifically capture states, too (e.g., the state-trait anxiety inventory was developed to assess situational and general dispositions; Spielberger, 2010).

3) Relatedly, the authors appear to presuppose that social trust is a stable personality disposition. But is trust not highly variable across different contexts, as a function of different personal interactions, etc.? In other words, is the expected variability in trust (as measured by the repeated games; hypothesis 1) simply considered random noise, or is this the result of a systematic mechanism? Overall a more detailed elaboration on how the authors conceptualize social trust would be helpful.

* Logic, rationale, and plausibility of the proposed hypotheses

The hypotheses are plausible and clearly specified.

* Soundness and feasibility of the methodology and analysis pipeline (including statistical power analysis where applicable)

Generally the empirical approach is sound, but I have a series of comments in this regard.

1) The exclusion criteria concerning "obvious random responses" should be specified more clearly.

2) Participant recruitment will be stopped "once the target sample is reached"; this requires that the specified quality-control / exclusion criteria will have to be monitored on the fly (otherwise the final sample will be small than the aspired sample size). Is this the case?

3) Is there any concern for anchoring effects, given that participants fill in the questionnaire in the first session? What about splitting the sample into two groups (a: questionnaires pre & post repeated assessments; b: questionnaire only post repeated assessments)?

4) (How) will it be ensured that participants play the games only every second day?

5) I was confused by the statement "In the 8 intermediate days, respondents only play the Trust game 16 times" (p. 8). Why "only"? They also play 16 rounds in the initial session, or do they play more (if so, please clarify)?

6) The authors plan to use ICCs to quantify the reliability of the economic games. Instead of computing 16 separate ICCs and then averaging them, I wondered whether one could implement a single model that treats the 16 games (or even the 4 x 4 predictors) as fixed effects, to thus directly extract one overall indicator of intraindividual variability?

7) Although the analysis plan is specified quite well, but the authors should spell out explicitly under which conditions (thresholds for statistical tests, etc.) they consider their hypotheses confirmed.

* Whether the clarity and degree of methodological detail would be sufficient to replicate exactly the proposed experimental procedures and analysis pipeline

Yes, but see points above.

* Whether the authors provide a sufficiently clear and detailed description of the methods to prevent undisclosed flexibility in the experimental procedures or analysis pipeline

Yes, but see points above.

* Whether the authors have considered sufficient outcome-neutral conditions (e.g. positive controls) for ensuring that the results obtained are able to test the stated hypotheses

Yes, but there is some ambiguity as to how "close" is "close enough" compared to the results of Mell et al. (p. 11); please specify.

References:

Frey, R. (2020). Decisions from experience: Competitive search and choice in kind and wicked environments. *Judgment and Decision Making*, 15(2), 282–303.

Hertwig, R., & Ortmann, A. (2001). Experimental practices in economics: A methodological challenge for psychologists? *Behavioral and Brain Sciences*, 24(03), 383–403.

Spielberger, C. D. (2010). State-Trait anxiety inventory. *The Corsini encyclopedia of psychology*.

Steiner, M., D., Seitz, F., & Frey, R. (in press). Through the window of my mind: Mapping information integration and the cognitive representations underlying self-reported risk preference. *Decision*. <https://doi.org/10.31234/osf.io/sa834>

Author's Response to Decision Letter for (RSOS-210213.R0)

See Appendix B.

RSOS-210213.R1 (Revision)

Review form: Reviewer 2

Do you have any ethical concerns with this paper?

Yes

Recommendation?

Accept with minor revision

Comments to the Author(s)

The authors have done a great job in revising their registered report. I do not see any major outstanding issues and only have a minor re-comment. Specifically, the authors write:

"Because the dependent variable is on a 9-point Likert scale, we will use a Poisson distribution." Although a Poisson distribution is potentially better suited as compared to the Gaussian dist., it still is not quite the right distribution for modeling Likert responses (i.e., unlike the Likert scales, Poisson distributions do not have an upper bound). The appropriate model would be an ordered logistic model and the authors may want to consider using it -- which, however, makes it substantially more intricate to interpret the respective estimates.

I wish the authors good luck with the data collection and look forward to reading the final manuscript.

Decision letter (RSOS-210213.R1)

Dear Dr Lettinga,

On behalf of the Editors, I am pleased to inform you that your Manuscript RSOS-210213.R1 entitled "Variability in repeated economic games; comparing Trust game decisions to other social trust measures" has been accepted in principle for publication in Royal Society Open Science subject to minor revision in accordance with the referee and editor suggestions. Please find their comments at the end of this email.

The reviewers and handling editors have recommended publication, but also suggest some minor revisions to your manuscript. Therefore, I invite you to respond to the comments and revise your manuscript.

Please you submit the revised version of your manuscript within 7 days (i.e. by the 07-May-2021). If you do not think you will be able to meet this date please let me know immediately.

When submitting your revised manuscript, you will be able to respond to the comments made by the referees and you should upload a file "Response to Referees". You can use this to document any changes you make to the original manuscript. In order to expedite the processing of the revised manuscript, please be as specific as possible in your response to the referees.

Full author guidelines can be found here <https://royalsocietypublishing.org/rsos/registered-reports>.

on behalf of Professor Chris Chambers (Subject Editor, Royal Society Open Science)
openscience@royalsociety.org

Associate Editor Comments to Author (Professor Chris Chambers):

Associate Editor: 1

Comments to the Author:

One of the previous Stage 1 reviewers was available to assess the revised manuscript. The assessment is positive, and there is just one minor methodological point remaining to address. Provided the authors can respond adequately to this concern through revision or rebuttal, Stage 1 in-principle acceptance should be forthcoming without requiring further in-depth review.

Reviewer comments to Author:

Reviewer: 2

Comments to the Author(s)

The authors have done a great job in revising their registered report. I do not see any major outstanding issues and only have a minor re-comment. Specifically, the authors write:

"Because the dependent variable is on a 9-point Likert scale, we will use a Poisson distribution." Although a Poisson distribution is potentially better suited as compared to the Gaussian dist., it still is not quite the right distribution for modeling Likert responses (i.e., unlike the Likert scales, Poisson distributions do not have an upper bound). The appropriate model would be an ordered logistic model and the authors may want to consider using it -- which, however, makes it substantially more intricate to interpret the respective estimates.

I wish the authors good luck with the data collection and look forward to reading the final manuscript.

Author's Response to Decision Letter for (RSOS-210213.R1)

See Appendix C.

Decision letter (RSOS-210213.R2)

Dear Dr Lettinga

On behalf of the Editor, I am pleased to inform you that your Manuscript RSOS-210213.R2 entitled "Variability in repeated economic games; comparing Trust game decisions to other social trust measures" has been accepted in principle for publication in Royal Society Open Science.

You may now progress to Stage 2 and complete the study as approved. Before commencing data collection we ask that you:

- 1) Update the journal office as to the anticipated completion date of your study.
- 2) Register your approved protocol on the Open Science Framework (<https://osf.io/>) or other recognised repository, either publicly or privately under embargo until submission of the Stage 2 manuscript. Please note that a time-stamped, independent registration of the protocol is mandatory under journal policy, and manuscripts that do not conform to this requirement cannot be considered at Stage 2. The protocol should be registered unchanged from its current approved state, with the time-stamp preceding implementation of the approved study design.

Following completion of your study, we invite you to resubmit your paper for peer review as a Stage 2 Registered Report. Please note that your manuscript can still be rejected for publication at Stage 2 if the Editors consider any of the following conditions to be met:

- The results were unable to test the authors' proposed hypotheses by failing to meet the approved outcome-neutral criteria.
- The authors altered the Introduction, rationale, or hypotheses, as approved in the Stage 1 submission.
- The authors failed to adhere closely to the registered experimental procedures. Please note that any deviations from the approved experimental procedures must be communicated to the editor immediately for approval, and prior to the completion of data collection. Failure to do so can result in revocation of in-principle acceptance and rejection at Stage 2 (see complete guidelines for further information).
- Any post-hoc (unregistered) analyses were either unjustified, insufficiently caveated, or overly dominant in shaping the authors' conclusions.
- The authors' conclusions were not justified given the data obtained.

We encourage you to read the complete guidelines for authors concerning Stage 2 submissions at <https://royalsocietypublishing.org/rsos/registered-reports#ReviewerGuideRegRep>. Please especially note the requirements for data sharing, reporting the URL of the independently

registered protocol, and that withdrawing your manuscript will result in publication of a Withdrawn Registration.

Once again, thank you for submitting your manuscript to Royal Society Open Science and we look forward to receiving your Stage 2 submission. If you have any questions at all, please do not hesitate to get in touch. We look forward to hearing from you shortly with the anticipated submission date for your stage two manuscript.

on behalf of Professor Chris Chambers (Registered Reports Editor, Royal Society Open Science)
openscience@royalsociety.org

Author's Response to Decision Letter for (RSOS-210213.R2)

See Appendix D.

Decision letter (RSOS-210213.R3)

Dear Dr Safra:

I am pleased to inform you that your manuscript entitled "Variability in repeated economic games: comparing Trust game decisions to other social trust measures" is now accepted for publication in Royal Society Open Science.

Please remember to make any data sets or code libraries 'live' prior to publication, and update any links as needed when you receive a proof to check - for instance, from a private 'for review' URL to a publicly accessible 'for publication' URL. It is also good practice to add data sets, code and other digital materials to your reference list.

Royal Society Open Science is a fully open access journal. A payment may be due before your article is published. Our partner Copyright Clearance Center's RightsLink for Scientific Communications will contact the corresponding author about your open access options from the email domain @copyright.com (if you have any queries regarding fees, please see <https://royalsocietypublishing.org/rsos/charges> or contact authorfees@royalsociety.org).

The proof of your paper will be available for review using the Royal Society online proofing system and you will receive details of how to access this in the near future from our production office (openscience_proofs@royalsociety.org). We aim to maintain rapid times to publication after acceptance of your manuscript and we would ask you to please contact both the production office

and editorial office if you are likely to be away from e-mail contact to minimise delays to publication. If you are going to be away, please nominate a co-author (if available) to manage the proofing process, and ensure they are copied into your email to the journal.

on behalf of Professor Chris Chambers (Subject Editor).

Follow Royal Society Publishing on Twitter: @RSocPublishing
Follow Royal Society Publishing on Facebook:
<https://www.facebook.com/RoyalSocietyPublishing/>
Read Royal Society Publishing's blog:
<https://royalsociety.org/blog/blogsearchpage/?category=Publishing>

Appendix A

Review for Royal Society Open Science:

Variability in repeated economic games; comparing
Trust game decisions to other social trust measures

Note

My overall impression is that this article could constitute a publishable piece of scholarship. However, as written, I have a few concerns, some major and some minor. Overall, the idea to compare game play to both self-report data and ‘real world’ data is a good one. However, the theory needs to be made more precise, the statistical tools and work-flow need to be improved, and the limitations of this study design should be more carefully described.

Comments

- The overall level of theoretical depth seems under-researched as written. For example, the second sentence of the abstract states that economic games are not predictive of real world behavior. Well which games specifically? RICH economic games (see [1] and [2, 3]) have been shown to have very high external validity; in the case of [2], behavior in experimental allocations almost perfectly mirrors real-world food/money lending networks, and in [3] ethnic parochialism in game allocations closely resembles ethnic parochialism in social network ties and food/money lending ties.

Sure, I agree that behavior in some economic games might not always be predictive of some real-world behavior. But, behavior in games is not *necessarily* supposed to mirror real-world behavior. Economic games can be designed to capture real-world preferences. Likewise, they can be designed to study ideal-world preferences, unencumbered by real-world constraints. I’d be more clear about this.

The authors should refer to specific games, and specific inferential goals, and not talk about ‘economic games’ as if they are all the same. More precision is needed.

- The third sentence of the abstract then claims that the lack of external validity of economic games could be due to the fact that decisions made in them are not stable over time. Again, which economic games? All of them? Second, even if this claim happens to be true, I see no reason why this would logically follow from the preceding sentence.

Game play can be stable or variable over time, and can either parallel real-world behavior or allow for expression of ideal-world preferences. It all depends on which game one is talking about, and which constraints a person finds themselves under at a given moment in time.

- I worry about the claim made in the sentence: “For example, Chuang and Schechter (2015) showed that Trust game decisions in 2002, Dictator game decisions in 2007 and 2009 and Reciprocity game decisions...”.

The fact that different games (which measure *different* constructs and were played were played years apart) are uncorrelated does not imply that preferences are not stable over time. Again, which preferences are being talked about here? Trust preferences? Bargaining preferences? Punishment preferences? The preferences elicited by a Trust game are not the necessarily the same as those elicited by a Dictator game.

- There is no real mention that “talk is cheap” and that self-report questionnaires cannot be accepted as truth either. There needs to be some citation here to literature that quantifies self-report biases. Respondents will often respond in ways that allow themselves to present themselves to themselves (and the researchers) in a positive light. If I say that I am generous now, and say that I am generous in the future, my consistent self-delusion of being generous (when I am not) is not an indication that I am really generous. It just means I am consistent in over-representing how generous I am. If you want see how generous I really am, then you have to see how generous I am in contexts where I actually stand

to gain or lose money from my behavior. This might come from game-play data or from observational data collection. The fact that self-reported trust measures don't correlate well with observed behavior (see [3]) should be a clue that self-report isn't always a defensible method of measuring preferences; we benefit from triangulating based on multiple measures, as the authors propose to do here.

The study that the authors propose here will be great, because we get to see: 1) what people do in an economic game, 2) what people say they would do, and 3) what people really do. So my comment here isn't meant to be discouraging. Conceptually, the paper is on the right track, I would just add that self-report data are likely to be biased in their own way too.

Also, I might consider the guidance in [2] that we should not use specific economic games or survey questions just because they are "standard" or "uncontroversial". The value of any research tool depends on the question it will be used to address. If the authors' question here involves comparing game-play behavior, self-reported preferences, and real trust behavior, I would modify the classic Trust questions and game setup a bit, so as to have them better measure the same underlying construct in different ways. Ask what people would do in a vignette that involves trust, and have them play a game that based on a similar setup.

- Wait, in the abstract the authors say that "economic games decisions are less stable over time". Then on page 6 they say "Lnnqvist et al. (2015) ... showed that two Trust game decisions made over a 2-year period were stable". This seems pretty weird, especially since the evidence they present for economic game decisions not being stable comes from a study comparing different games in different years (i.e., Chuang & Schechter, 2015).

This seems scrambled. I would revise the argument substantially. If the study using the *same game in two time-points* shows that game play is stable over time, then don't lead off the abstract with the opposite claim based on a weaker study design.

- There are some worries I have about the analysis plan. Montgomery et al. [4] have a paper called "How conditioning on post-treatment variables can ruin your experiment," where they address causal inference problems introduced by common methodological choices that are also described in the authors' research plan. I would check this paper out before doing things like conditioning the sample on post-treatment variables (i.e., dropping individuals for responding consistently in a study that tests for consistency of responses).
- The following might be a sensitive topic, and I don't mean to sound accusatory, but I worry about the use of deceit in the study design. This might be a disciplinary distinction, but I would never lie to my research participants about the research methods they will be subjected to. Participants cannot provide informed consent to participate in research if they are not informed—or, in this case, if they are specifically misled—about what is actually being done. Especially if the research is based around actually receiving a financial payout, some might consider it fraudulent to claim that one thing will happen: "...the respondents will be told that the role of the trustee will be played by a different person each day," when in fact something else is happening: "...in order to accurately randomize the reciprocation probability, the role of the trustee is simulated by a computer program". If participants weren't deceived, they might act differently, and this has financial consequences for them. I, personally, would be upset if I was a participant and found out later that I was lied to about the study procedures determining my earnings.

If an official ethics board has approved this study design, it is probably not my role as a reviewer to impose my own beliefs here. This is a decision that the authors and publisher have to make on their own.

It is worth considering if deceit is scientifically necessary, or scientifically harmful. As a player, if a researcher told me that there are specific probabilities of reciprocation, *and* that I was playing with a real player, I would know that they are lying to me somehow, and I wouldn't take the game seriously.

Instead of using deceit, and playing these game with a ‘convenience sample’ online, and then dropping people based on their answers being too consistent (as an ‘attention check’), couldn’t the science be improved by playing the games with local community members, in-person, without using deceit?

I would advise actually playing the Trust game, with real people on both ends, just like you tell the respondents you are doing. Then adopt statistical methods that are appropriate for analyzing that kind of data (e.g., see the Social Relations Model [5, 2, 6] for dealing with sender, receiver, and dyadic effects).

- I would check out a 21-st century statistics textbook (e.g., McElreath [6] or Gelman et al. [7]) before doing some of the practices stated in the methods section.

Statements like: “The assumption of normal distribution will be visually inspected for the willingness-to-play..” don’t make much sense, because the data generating process here does not produce Gaussian data. The outcomes are ordered-categorical data, so they cannot possibly have a normal distribution. Normally distributed data do not pile up on integer values between 1 and 9. Given that the outcomes are ordered-categorical data, they should be modeled as such statistically. Pearson tests, and everything else based on normally distributed data, are not the right tools here.

I’d recommend browsing McElreath’s [6] chapters on ordered categorical data. Then, make up a script in R that generates simulated ordered categorical data with temporal correlations at the individual level. Then, write up the data analysis script in R or Stan, and check that your model allows you to recover the correct parameter estimates. This script should be included the with preregistration. This way, a reviewer can carefully evaluate your methods.

- Think about writing up a generative model. For the first analysis, I’d use something like this: Let $Y_{[i,t,q]}$ be the ordered categorical response by individual i at time point $t \in \{1, 2\}$ with regard to question q (currently, the authors have 16 of these questions based on the interaction of total coins and reciprocation probability).

$$Y_{[i,j,q]} \sim \text{Ordered Categorical}(\alpha + \beta_{[q]} + \gamma_{[i,t]}, C) \tag{1}$$

This says that the outcome is a function of an intercept, α , a question-type random effect, β , an individual-specific random effect, γ , and a vector of cutpoints, C . The β parameter is modeled hierarchically as:

$$\beta_{[q]} \sim \text{Normal}(0, \sigma) \tag{2}$$

Assuming that we are looking at just two time points, we could then model γ as:

$$\gamma_{[i,1:2]} \sim \text{Multivariate Normal}(Z, \varsigma\Gamma\varsigma) \tag{3}$$

where Z is a 2-vector of zeroes, ς is a diagonal matrix of standard deviation terms, and Γ is a correlation matrix. The off-diagonal of Γ tells us how correlated an individual’s responses are across time, after accounting for question effects, and the non-Gaussian nature of the outcomes.

- There are a lot of other weird things, like the removal of ‘outliers’, that I find rather shocking. I didn’t know people still did this. If the data generating process produces “outliers”, then removing these outliers prevents you from fully understanding the data generating process. Instead, find an analytical framework suited to your data set, rather than manipulating the data to suite an inappropriate statistical framework. If you do this analysis with the right tools, you won’t have to do the *ad hoc* things.
- Dropping data points if the player responds the same way each time is also stunning. So if somebody is consistently selfish, on all 16 trials, in both time points, and never gives coins to the other person,

then the authors will drop this person from the analysis? If the authors are going to drop players for responding consistently, it is really hard to imagine how they can measure if people are consistent over time!

- Finally, I restate my prior worry about using a simple online tool to collect data. Does the research team really believe that this online platform provides the best way to get an unbiased sample of respondents which allows them to generalize their findings to inform our knowledge of human psychology? Or is this a simple, low effort way to get “data”. I have my doubts that this is the best way to conduct a rigorous study. For the last 15 years or more, scientists who conduct field research have been pushing for better sampling designs, less reliance on convenience samples, and more focus on robust, comparative data collection [8, 9, 10, 11]; they might have some ideas worth taking to heart that will only improve the power and generalizability of this study.

Good work so far, and best wishes on the manuscript.

References

- [1] Matthew M Gervais. Rich economic games for networked relationships and communities: development and preliminary validation in yasawa, fiji. *Field methods*, 29(2):113–129, 2017.
- [2] Anne C Pisor, Matthew M Gervais, Benjamin G Purzycki, and Cody T Ross. Preferences and constraints: the value of economic games for studying human behaviour. *Royal Society open science*, 7(6): 192090.
- [3] Anne Pisor and Cody T Ross. How generalizable are patterns of parochial altruism in humans? URL <https://osf.io/tc7xa/download>.
- [4] Jacob M Montgomery, Brendan Nyhan, and Michelle Torres. How conditioning on posttreatment variables can ruin your experiment and what to do about it. *American Journal of Political Science*, 62(3): 760–775, 2018.
- [5] Jeremy M Koster and George Leckie. Food sharing networks in lowland Nicaragua: An application of the social relations model to count data. *Social Networks*, 38:100–110, 2014.
- [6] Richard McElreath. *Statistical rethinking: A Bayesian course with examples in R and Stan*. CRC press, 2020.
- [7] Andrew Gelman, John B Carlin, Hal S Stern, David B Dunson, Aki Vehtari, and Donald B Rubin. *Bayesian data analysis*. CRC press, 2013.
- [8] Joseph Henrich, Robert Boyd, Samuel Bowles, Colin Camerer, Ernst Fehr, Herbert Gintis, Richard McElreath, Michael Alvard, Abigail Barr, Jean Ensminger, et al. economic man in cross-cultural perspective: Behavioral experiments in 15 small-scale societies. *Behavioral and brain sciences*, 28(6): 795–855, 2005.
- [9] Paul Rozin. What kind of empirical research should we publish, fund, and reward?: A different perspective. *Perspectives on Psychological Science*, 4(4):435–439, 2009.
- [10] Joseph Henrich, Steven J Heine, and Ara Norenzayan. The weirdest people in the world? *Behavioral and brain sciences*, 33(2-3):61–83, 2010.
- [11] Leonid Tiokhin, Joseph Hackman, Shirajum Munira, Khaleda Jesmin, and Daniel Hruschka. Generalizability is not optional: insights from a cross-cultural study of social discounting. *Royal Society open science*, 6(2):181386, 2019.

Appendix B

Subject: Resubmitted Manuscript RSOS-210213

Dear Pr. Chambers,

Thank you for your helpful review of our manuscript and for providing us with the opportunity to revise and resubmit our work. We are grateful that you accepted our request to extend the resubmission deadline by a few days. We wish to thank the reviewers for the positive evaluation, helpful comments and advice. Below we provide a point-by-point reply to the reviewers' comments, with changes to the manuscript located below each point. We addressed all the reviewers' comments and made significant changes to the manuscript.

We hope you will find this version of the manuscript acceptable.

With kind regards,

Niels Lettinga, Lou Safra, Pierre O. Jacquet, Coralie Chevallier

Editor

Two expert reviewers have now assessed the Stage 1 manuscript and have provided very constructive and detailed reviews. Both reviewers find merit in the proposal while also highlighting a wide range of issues that will need to be addressed to progress the manuscript toward in-principle acceptance. Headline issues include the theoretical framing of the research (and level of theoretical depth), concerns with the deception manipulation (both ethical and also scientific in terms of its effectiveness), concerns with self-report measures, the validity of the proposed analysis pipeline, and the clear linking of the predictions with the analyses that will confirm or disconfirm them. The reviews also contain many detailed comments concerning the proposed methods and the need for greater clarification and justification of design decisions.

Reviews this critical would likely lead to rejection of a completed study, but since the RR process gives authors to opportunity to address serious concerns before research is undertaken, I would like to offer the authors the opportunity to submit a Major Revision. Any revised manuscript will be returned to the reviewers for reassessment.

Reviewer 1

Note

My overall impression is that this article could constitute a publishable piece of scholarship. However, as written, I have a few concerns, some major and some minor. Overall, the idea to compare game play to both self-report data and `real world' data is a good one. However, the theory needs to be made more precise, the statistical tools and work-flow need to be improved, and the limitations of this study design should be more carefully described.

We thank the reviewer for the positive evaluation of the work and appreciate the insightful comments. We have integrated all of them in our new version, which we believe greatly improved the manuscript.

Comments

The overall level of theoretical depth seems under-researched as written. For example, the second sentence of the abstract states that economic games are not predictive of real world behavior. Well which games specifically? RICH economic games (see Gervais (2017) and (Pisor et al., 2020; Pisor & Ross, 2021)) have been shown to have very high external validity; in the case of Pisor et al. (2020), behavior in experimental allocations almost perfectly mirrors real-world food/money lending networks, and in Pisor and Ross (2021) ethnic parochialism in game allocations closely resembles ethnic parochialism in social network ties and food/money lending ties.

Sure, I agree that behavior in some economic games might not always be predictive of some real-world behavior. But, behavior in games is not necessarily supposed to mirror real-world behavior. Economic games can be designed to capture real-world preferences. Likewise, they can be designed to study ideal-world preferences, unencumbered by real-world constraints. I'd be more clear about this.

The authors should refer to specific games, and specific inferential goals, and not talk about `economic games' as if they are all the same. More precision is needed.

The reviewer is right that our presentation of the lack of external validity of economic games needed to be nuanced. It is indeed true that some economic games are predictive of real-world behavior and that the use of economic games is not restricted to predicting real-world behavior. We base our argumentation primarily on a recent meta-analysis and lab-field experiment by Galizzi and Navarro-Martinez (2019), which focused on four economic games: the Dictator game, the Ultimatum game, the Public Goods game, and most importantly for our purposes, the Trust game. The authors found that these four economic games, which cover a large fraction of experimental research on social preferences, lack external validity. We apologize for the over-generalization in the original manuscript, and we now mention these four specific economic games instead.

We added the aforementioned details in the abstract and on p.3 of the revised manuscript.

Page 3: "Galizzi and Navarro-Martinez' (2019) work focused on the Dictator game, the Ultimatum game, the Public Goods game and the Trust game and it is possible that other economic games are more predictive of real-world behavior. Nonetheless, the four economic games analyzed by Galizzi and Navarro-Martinez (2019) cover a large fraction of experimental research on social preferences. In what follows, we mostly refer to evidence pointing to a lack of external validity of these standard four economic games."

The third sentence of the abstract then claims that the lack of external validity of economic games could be due to the fact that decisions made in them are not stable over time. Again, which economic games? All of them? Second, even if this claim happens to be true, I see no reason why this would logically follow from the preceding sentence.

Game play can be stable or variable over time, and can either parallel real-world behavior or allow for expression of ideal-world preferences. It all depends on which game one is talking about, and which constraints a person finds themselves under at a given moment in time.

As mentioned in the previous comment, regarding the lack of external validity of economic games, we are specifically referring to four economic games: the Dictator game, the Ultimatum game, the Public Goods game and the Trust game. More specifically, we are only investigating the stability of the Trust game. The reason for this is that in a recent study by Lettinga et al. (2021), the correlation between Trust game decisions and self-reported social trust was almost null ($\rho = 0.00$, $p = 0.92$), using a large sample ($N = 612$). This in line with other studies, that showed that Trust game decisions and questionnaires that measure social trust are not correlated (Galizzi & Navarro-Martinez, 2019; McAuliffe et al., 2019).

It is true that the fact that economic games lack external validity does not necessarily imply that this comes from their instability. One of the goals of our paper is in fact to test whether temporal instability plays a role. As mentioned in the manuscript, the only direct evidence of the stability of Trust game decisions comes from Lönnqvist et al. (2015), who showed that two Trust game decisions made by the trustor over a 1-year period were stable. However, their sample size was very small ($N = 22$), which makes the evidence rather inconclusive.

Based on the reviewer's comment, we have made several changes in the abstract of the revised manuscript.

Abstract: "Economic games are well-established tools that offer a convenient approach to study social behavior. Although economic games are widely used, recent evidence suggests that decisions made in the context of standard economic games (i.e., Ultimatum game, Dictator game, Public Goods game and Trust game) are less predictive of real-world behavior than previously assumed and than self-reported questionnaires. A possible explanation for this discrepancy is that economic games decisions in the lab are more likely to be influenced by the current situation (i.e., participants' current state), while questionnaires are specifically designed to measure people's average behavior across a long period of time (i.e., a stable trait). If this explanation is correct, then we should see that repeating economic games over an extended period of time should bring participants closer to the average behavior that questionnaires capture. To test this hypothesis, we will perform a longitudinal study where 275 respondents will play 16 Trust games per day for 10 days total within a 3-week period, and fill out a questionnaire that measures social trust."

I worry about the claim made in the sentence: "For example, Chuang and Schechter (2015) showed that Trust game decisions in 2002, Dictator game decisions in 2007 and 2009 and Reciprocity game decisions...".

The fact that different games (which measure different constructs and were played were played years apart) are uncorrelated does not imply that preferences are not stable over time. Again, which preferences are being talked about here? Trust preferences? Bargaining preferences? Punishment preferences? The preferences elicited by a Trust game are not the necessarily the same as those elicited by a Dictator game.

We agree with the reviewer that the design used by Chuang and Schechter (2015) doesn't measure the stability of a specific economic game. We decided to not mention this study in the manuscript anymore (NB: Chuang and Schechter (2015) also review the literature on the stability of economic games and self-reported questionnaires measuring social, risk and time preferences, which we still refer to in the manuscript). In another study, Brosig et al. (2007) found that stability in various Dictator and Prisoner's Dilemma games over a 3-month period was low, but their sample size was small ($N = 40$). In this study, stable behavior was only

found for participants who behaved consistently selfishly. Participants who behaved prosocially in the beginning of the experiment tend to behave more selfishly as the experiment unfolds.

We have added the aforementioned details on p.3 and on p.4 of the revised manuscript.

Page 3 and 4: “For example, using a design where participants played the same games over a 3-month period (i.e., various Dictator and Prisoner’s Dilemma games), Brosig et al. (2007) found that stability was low while Lönnqvist et al. (2015) found that decisions in the Trust game were stable over a 1-year period. Unfortunately, both of these studies relied on very small samples (N = 40 and N = 22 for Brosig et al. (2007) and Lönnqvist et al. (2015) respectively), which considerably limits the generalizability of the results.”

There is no real mention that “talk is cheap” and that self-report questionnaires cannot be accepted as truth either. There needs to be some citation here to literature that quantifies self-report biases. Respondents will often respond in ways that allow themselves to present themselves to themselves (and the researchers) in a positive light. If I say that I am generous now, and say that I am generous in the future, my consistent self-delusion of being generous (when I am not) is not an indication that I am really generous. It just means I am consistent in over-representing how generous I am. If you want see how generous I really am, then you have to see how generous I am in contexts where I actually stand to gain or lose money from my behavior. This might come from game-play data or from observational data collection. The fact that self reported trust measures don't correlate well with observed behavior (see Pisor & Ross (2021)) should be a clue that self-report isn't always a defensible method of measuring preferences; we benefit from triangulating based on multiple measures, as the authors propose to do here.

The study that the authors propose here will be great, because we get to see: 1) what people do in an economic game, 2) what people say they would do, and 3) what people really do. So my comment here isn't meant to be discouraging. Conceptually, the paper is on the right track, I would just add that self-report data are likely to be biased in their own way too.

Also, I might consider the guidance in Pisor et al. (2020) that we should not use specific economic games or survey questions just because they are “standard” or “uncontroversial”. The value of any research tool depends on the question it will be used to address. If the authors' question here involves comparing game-play behavior, self-reported preferences, and real trust behavior, I would modify the classic Trust questions and game setup a bit, so as to have them better measure the same underlying construct in different ways. Ask what people would do in a vignette that involves trust, and have them play a game that based on a similar setup.

The reviewer is right that self-reported questionnaires have their own limitations. First, they can be biased by social desirability, especially in the context of prosocial behavior (Krumpal, 2013). Second, it has been demonstrated that people have surprisingly limited insight on their mental states and preferences (Kolar et al., 1996). Thus, even when people try to accurately predict their own behavior, they might not be able to do so. Third, even if people accurately describe their own attitudes, this might not be correlated with their real behavior because other factors might affect behavioral decisions (Ajzen, 2005). We have added the aforementioned details on p.4 of the revised manuscript.

As the reviewer states, the fact that we measure social trust via three different methods (self-reported questionnaire, Trust game and real-world behavior) should allow us to determine how well self-reported behavior measured via a questionnaire correlates with real-world behavior related to social trust.

Finally, we agree that the value of any research tool depends on the question that it seeks to address. However, because we are interested in whether repeating a Trust game

increases its correlation with questionnaire responses or real-world behavior; we chose to use a standard Trust game. This does not mean that other ways to increase external validity, such as adjusting the characteristics of the games so that they are more aligned with the specific social behavior, are not interesting (Goeschl et al., 2015; Pisor et al., 2020). Rather, this is not what we are trying to investigate in this particular study. It will be interesting to incorporate these elements in the discussion section of the manuscript.

Page 4: “Although, self-reported questionnaires have their own limitations: people can be biased by social desirability (Krumpal, 2013) and sometimes have limited insight on their mental states and preferences (Kolar et al., 1996).”

Wait, in the abstract the authors say that “economic games decisions are less stable over time”. Then on page 6 they say “Lönqvist et al. (2015) ... showed that two Trust game decisions made over a 2-year period were stable”. This seems pretty weird, especially since the evidence they present for economic game decisions not being stable comes from a study comparing different games in different years (i.e., Chuang & Schechter, 2015).

This seems scrambled. I would revise the argument substantially. If the study using the same game in two time-points shows that game play is stable over time, then don't lead off the abstract with the opposite claim based on a weaker study design.

We completely agree with the reviewer that our original argumentation was not sufficiently clear. As mentioned previously, we decided to not mention the study of Chuang and Schechter (2015) in the manuscript anymore (NB: Chuang and Schechter (2015) also review the literature on the stability of economic games and self-reported questionnaires measuring social, risk and time preferences, which we still refer to in the manuscript). As mentioned in a previous comment, Brosig et al. (2007) found that behavioral stability during various Dictator and Prisoner's Dilemma games over a 3-month period was low, but their sample size was small (N = 40). Besides this instability of social preferences, instability has also been shown for other behavioral tasks measuring, for example, risk preferences (Frey et al., 2017).

The only direct evidence about the stability of Trust game decisions comes from Lönqvist et al. (2015), who showed that two Trust game decisions made by the trustor over a 1-year period were stable, but their sample size was very small (N = 22). Whether or not Trust game decisions are stable over time therefore remains an open question, which our paper will contribute to with a large pool of participants.

We have added the aforementioned details on p.3 and on p.4 of the revised manuscript.

Page 3 and 4: “For example, using a design where participants played the same games over a 3-month period (i.e., various Dictator and Prisoner's Dilemma games), Brosig et al. (2007) found that stability was low while Lönqvist et al. (2015) found that decisions in the Trust game were stable over a 1-year period. Unfortunately, both of these studies relied on very small samples (N = 40 and N = 22 for Brosig et al. (2007) and Lönqvist et al. (2015) respectively), which considerably limits the generalizability of the results.”

The following might be a sensitive topic, and I don't mean to sound accusatory, but I worry about the use of deceit in the study design. This might be a disciplinary distinction, but I would never lie to my research participants about the research methods they will be subjected to. Participants cannot provide informed consent to participate in research if they are not informed-or, in this case, if they are specifically mislead-about what is actually being done. Especially if the research is based around actually receiving a financial payout, some might consider it fraudulent to claim that one thing will happen: “...the respondents will be told that the role of the trustee will be played by a different person each day,” when in fact something else is happening: “...in order to accurately randomize the reciprocation probability, the role of the trustee is simulated by a computer program”. If participants weren't

deceived, they might act differently, and this has financial consequences for them. I, personally, would be upset if I was a participant and found out later that I was lied to about the study procedures determining my earnings.

If an official ethics board has approved this study design, it is probably not my role as a reviewer to impose my own beliefs here. This is a decision that the authors and publisher have to make on their own.

It is worth considering if deceit is scientifically necessary, or scientifically harmful. As a player, if a researcher told me that there are specific probabilities of reciprocation, and that I was playing with a real player, I would know that they are lying to me somehow, and I wouldn't take the game seriously. Instead of using deceit, and playing these game with a `convenience sample' online, and then dropping people based on their answers being too consistent (as an `attention check'), couldn't the science be improved by playing the games with local community members, in-person, without using deceit?

I would advise actually playing the Trust game, with real people on both ends, just like you tell the respondents you are doing. Then adopt statistical methods that are appropriate for analyzing that kind of data (e.g., see the Social Relations Model (Koster & Leckie, 2014; Pisor et al., 2020; McElreath, 2020) for dealing with sender, receiver, and dyadic effects).

We thank the reviewer for highlighting this issue. We agree that participants might figure out that they are not playing with a real and different partner in each game, which could cause participants to behave in an unrealistic manner. In addition, we agree that deceit isn't necessary here. We therefore decided to change the instructions to provide more transparent information. Specifically, we will inform participants that they will play with "virtual partners", and that each game is played with a different virtual partner who behaves independently. Participants will be informed that these virtual partners are not real people but are programs that can simulate real-life behavior, just like characters in videogames. This removes the deceptive element and allows us to keep control over the reciprocation probabilities. Importantly, recent work shows that people invest similarly in humans and robots in a Trust game where the participants knew their partner was a robot or human (Schniter et al., 2020).

We have added the aforementioned details on p.9 of the revised manuscript.

Page 9: "The respondents will be told that they will play with "virtual partners", and that each game is played with a different virtual partner (this counteracts any reputation effects) who behaves independently. Respondents will be informed that these virtual partners are not real people but are programs that can simulate real-life behavior, just like characters in videogames. Importantly, recent work shows that people invest similarly in humans and robots in a Trust game where the participants knew their partner was a robot or human (Schniter et al., 2020)."

The reviewer's comments regarding "playing these games with a `convenience sample' online" and "dropping people based on their answers being too consistent (as an `attention check')" will be discussed in subsequent comments.

There are some worries I have about the analysis plan. Montgomery et al. (2018) have a paper called "How conditioning on post-treatment variables can ruin your experiment," where they address causal inference problems introduced by common methodological choices that are also described in the authors' research plan. I would check this paper out before doing things like conditioning the sample on post- treatment variables (i.e., dropping individuals for responding consistently in a study that tests for consistency of responses).

We thank the reviewer for highlighting these issues. We are in full agreement with Montgomery et al.'s (2018) recommendations. Montgomery et al. (2018) mention several issues that can cause posttreatment bias, such as "dropping participants who fail posttreatment manipulation checks" and "including posttreatment variables as covariates" in

the context of randomized experiments (where one group receives a treatment and the other group does not). However, this does not apply to our case since we do not divide our sample into different treatment groups. Nonetheless, we removed the exclusion criterion regarding “obvious random responses” in response to another comment provided by the reviewer. The sentence concerning the exclusion criteria on p.7 of the revised manuscript is now as follows:

Page 7: “Respondents will be excluded when they do not complete the entire experiment, when they do not provide correct answers during the last of the two comprehension check sections on the first and last day of the experiment, when their reaction time is below 200ms on at least 90% of the games, or when multiple respondents use the same IP address.”

I would check out a 21-st century statistics textbook (e.g., McElreath (2020) or Gelman et al. 2013) before doing some of the practices stated in the methods section.

Statements like: “The assumption of normal distribution will be visually inspected for the willingness- to-play..” don’t make much sense, because the data generating process here does not produce Gaussian data. The outcomes are ordered-categorical data, so they cannot possibly have a normal distribution. Normally distributed data do not pile up on integer values between 1 and 9. Given that the outcomes are ordered-categorical data, they should be modeled as such statistically. Pearson tests, and everything else based on normally distributed data, are not the right tools here.

I’d recommend browsing McElreath’s (2020) chapters on ordered categorical data. Then, make up a script in R that generates simulated ordered categorical data with temporal correlations at the individual level. Then, write up the data analysis script in R or Stan, and check that your model allows you to recover the correct parameter estimates. This script should be included the with preregistration. This way, a reviewer can carefully evaluate your methods.

Think about writing up a generative model. For the first analysis, I’d use something like this: Let $Y_{[i, t, q]}$ be the ordered categorical response by individual i at time point $t \in \{1, 2\}$ with regard to question q (currently, the authors have 16 of these questions based on the interaction of total coins and reciprocation probability).

$$Y_{[i, j, q]} \sim \text{Ordered Categorical}(\alpha + \beta_{[q]} + [i, t], C) \quad (1)$$

This says that the outcome is a function of an intercept, α , a question-type random effect, β , an individual-specific random effect, y , and a vector of cutpoints, C . The β parameter is modeled hierarchically as:

$$\beta_{[q]} \sim \text{Normal}(0, \sigma) \quad (2)$$

Assuming that we are looking at just two time points, we could then model y as:

$$y_{[i, 1:2]} \sim \text{Multivariate Normal}(Z, \zeta \Gamma \zeta) \quad (3)$$

where Z is a 2-vector of zeroes, ζ is a diagonal matrix of standard deviation terms, and Γ is a correlation matrix. The off-diagonal of Γ tells us how correlated an individual’s responses are across time, after accounting for question effects, and the non-Gaussian nature of the outcomes.

We adopted our methodology from Mell et al. (2021), who published a similar study with a Trust game played 16 times in a single session. In our study, we will use the same Trust game, but participants will play this game 16 times on 10 days. Our goal is to test the effect of this repetition, which led us to match our analytic strategy as closely as possible to that of Mell et al. (2021). We believe that putting a particular emphasis on replicability and generalizability is a goal that is worth pursuing.

The reviewer is right that “willingness-to-play” can be classified as ordered categorical, since it is measured on a 9-point Likert scale, but we would like to note that ordinal data extracted from responses on Likert scales with 5 or more points share many properties with continuous data and can be treated as such (Johnson & Creech, 1983; Norman, 2010; Sullivan & Artino, 2013; Zumbo & Zimmerman, 1993, we now cite these references in the manuscript). More importantly, our dependent variable represents an *average* behavior, whether measured at the end of day one (on the 16 games), or over all days (on the 16 games * 10 days), which means that our dependent variable closely approximates the properties of a continuous variable. In fact, the data collected in Mell et al. (2021) demonstrate that average willingness-to-play based on the 16 Trust games and calculated for 75 subjects consists of 46 different values ranging from 1 to 9, and whose distribution does not significantly deviate from normality (Shapiro-Wilk test: $W = 0.98$, $p = 0.31$).

We have added this information on p.14 and p.15 of the revised manuscript.

Page 14 and 15: “The willingness-to-play averaged on the first day and over all days is the dependent variable, labeled AWP (Average Willingness to Play). The single responses that give rise to AWP are measured via a 9-point Likert scale, and in principle could be classified as ordered categorical. However, AWP represents the average of these responses (16 on day 1; 160 over all days) and therefore includes a number of ranks that is large enough to consider it as approximately continuous. Note in addition, that responses measured on Likert scales with five or more points share many properties with continuous data and can be treated as such (Johnson & Creech, 1983; Norman, 2010; Sullivan & Artino, 2013; Zumbo & Zimmerman, 1993).”

The reviewer’s suggestion to analyze the participants’ responses on a trial-by-trial basis using Bayesian statistics is interesting, but we believe it is beyond the scope of the present study. Our goal with this paper is not to test a decision-making model, which would indeed require simulating fake participants who would play our games using a specific decision rule (e.g., soft max). Our goal is to focus on participants’ average behavior in the Trust game and test whether it correlates with trust questionnaire responses. If this is a promising area of research, future studies might include models that integrate individual trial-by-trial variability, or more advanced methods modelling some of the computations accounting for participants’ choices and their stochasticity along the task (e.g., Jacquet et al., 2019; Safra et al., 2019). This will require putting forward specific hypotheses about underlying decision-making models.

There are a lot of other weird things, like the removal of `outliers', that I find rather shocking. I didn't know people still did this. If the data generating process produces “outliers”, then removing these outliers prevents you from fully understanding the data generating process. Instead, find an analytical framework suited to your data set, rather than manipulating the data to suite an inappropriate statistical framework. If you do this analysis with the right tools, you won't have to do the ad hoc things.

We took the reviewer’s suggestion into account and will not exclude any outliers or outlier data points.

Dropping data points if the player responds the same way each time is also stunning. So if somebody is consistently selfish, on all 16 trials, in both time points, and never gives coins to the other person, then the authors will drop this person from the analysis? If the authors are going to drop players for responding consistently, it is really hard to imagine how they can measure if people are consistent over time!

We also removed this exclusion criterion.

Finally, I restate my prior worry about using a simple online tool to collect data. Does the research team really believe that this online platform provides the best way to get an unbiased sample of respondents which allows them to generalize their findings to inform our knowledge of human psychology? Or is this a simple, low effort way to get “data”. I have my doubts that this is the best way to conduct a rigorous study. For the last 15 years or more, scientists who conduct field research have been pushing for better sampling designs, less reliance on convenience samples, and more focus on robust, comparative data collection (Henrich et al., 2005; Rozin, 2009; Henrich et al., 2010; Tiokhin et al., 2019) they might have some ideas worth taking to heart that will only improve the power and generalizability of this study.

We agree with the reviewer that an online sample could be biased. However, the same goes for an in-person sample in the lab. Generally speaking, lab experiments conducted at the heart of high-income capital cities attract biased samples. This problem is potentially amplified because our study would require participants to come to the lab 10 times within a 3-week period. Given the design of our study, an onsite study might end up attracting a biased sample of students or people who have much free time. We therefore chose to conduct an online study precisely because this will allow us to reach a wider and more diverse pool of participants.

In addition, it has been impossible to run lab-based studies for the past year in our university and since we live in an area where Covid-19 has hit hard, university closures are unlikely to be lifted for the foreseeable future. If the only solution to make progress with our research was to run lab-based studies, we would of course wait. But running online work is in fact a better methodology to answer our question and not “a low effort way to get “data””. Further research should test to what extent it replicates in different populations tested in the field.

Good work so far, and best wishes on the manuscript.

Reviewer 2

Comments to the Author(s)

This is an interesting and well worked-out study registration: The research questions are clear and topical, the proposed design in principle make sense (but see below), and the analysis plan is mostly solid. In what follows I will list a series of comments and questions that hopefully will be useful to further strengthen this study. Most of these are relatively minor comments, but one comment is more fundamental and requires particular attention, which is why I raise this issue at the outset.

We are grateful to the reviewer for the positive evaluation of the work, and for having contributed with his/her comments and suggestions to significantly improve the quality of the manuscript.

Specifically, the entire study is about measuring social trust – but the authors essentially fool their participants by telling them that they will play each game "on the fly" with another real participant (who is actually even supposed to be substituted with another real participant from trial to trial!). In reality, however, it is only the computer playing with the actual participants. The "data-sharing test" in the end is another deception, as participants who agree to share their data will be told that "they were not selected" (and in reality the data of nobody will ever be shared). I understand that there are some intricacies when studying social trust, but with the current implementation I see the real danger that participants will not buy the story that they are indeed playing with real (and different) partners on the fly (e.g., they will never have to wait for the response of the other player, making it quite easy to detect that some deception is going on). Thus, assuming that participants realize that they were tricked and actually play with a machine, why should they themselves trust the experimenters / this machine? And thus, can one indeed expect that participants will ultimately behave consistently / realistically? Importantly, there exist various internet forums in which participants quickly exchange insights gained in such online studies; hence there may be backfiring effects for this experiment specifically, but also for similar future studies more generally (see Hertwig & Ortmann, 2001). With current technologies it would be possible, in principle, to set up a study on social trust without any deception, including on platforms such as mTurk or Prolific (e.g., for a truly interactive online game see Frey, 2020). All in all, this issue definitely requires some further deliberation and potentially the consideration of alternative empirical approaches.

We thank the reviewer for highlighting this issue. We agree that participants might figure out that they are not playing with a real and different partner in each game, which could cause participants to behave in an unrealistic manner. In addition, we agree that deceit isn't necessary here. We therefore decided to change the instructions to provide more transparent information. Specifically, we will inform participants that they will play with "virtual partners", and that each game is played with a different virtual partner who behaves independently. Participants will be informed that these virtual partners are not real people but are programs that can simulate real-life behavior, just like characters in videogames. This removes the deceptive element and allows us to keep control over the reciprocation probabilities. Importantly, recent work shows that people invest similarly in humans and robots in a Trust game where the participants knew their partner was a robot or human (Schniter et al., 2020).

We have added the aforementioned details on p.9 of the revised manuscript.

Page 9: "The respondents will be told that they will play with "virtual partners", and that each game is played with a different virtual partner (this counteracts any reputation effects) who behaves independently. Respondents will be informed that these virtual partners are not real people but are programs that can simulate real-life behavior, just like characters in videogames. Importantly, recent work shows that

people invest similarly in humans and robots in a Trust game where the participants knew their partner was a robot or human (Schniter et al., 2020).”

Regarding the deception used in the data sharing scenario, we again agree with the reviewer’s suggestion. We decided to change the scenario substantially. We now present the participants first with an extra set of 13 items about their personal life at the end of day 10 (the last day of the experiment). After that, we will show them the scenario presented below and ask them if they agree (yes/ no) that their answers to these questions will be linked to the survey data collected during the past 10 days and will be shared with other researchers by uploading their anonymized data to the Open Science Framework (OSF). Only for the participants who select “yes” will their data be shared online. Therefore, no deception is used in this scenario anymore. The 13 additional questions and the new scenario and can be found on p.10 to p.12 of the revised manuscript.

Page 10 to 12: “Data sharing A new scenario proposed by Bauer et al. (2019) to measure real-world behavior related to social trust is whether respondents share, i.e., entrust their data to others. Specifically, they ask respondents for their permission to include additional external private data (i.e., data from social insurance carriers) to be included in the analysis. Respondents are therefore asked to trust the researchers with this additional data and to not abuse it, making it straightforwardly related to social trust. Here we adopt a similar strategy. At the end of the experiment on day 10 (after the Trust games are played and the three trust items are filled out), all respondents will be presented with a set of 13 additional questions about their personal life. This questionnaire will be entitled “Personal information about your past and current life”, and will be introduced as follows:

In the final part of this experiment, you will be presented with 13 additional questions about your personal life. These questions relate to the way your life was going on at home during childhood, as well as your current life, such as your financial situation and your health status.

Items 1 and 2 inform about the respondents gender (item 1) and age in years (item 2). Items 3 to 8 are taken from the well-established questionnaire designed by Mittal et al. (2015), and which assess the life conditions that respondents experienced during their childhood. These items inform about the socioeconomic status of the family household and the unpredictability of the family environment of the respondents when they were younger than 10 years of age. The following instructions will first be presented: “Think back to your life when you were younger than ten. This time includes preschool, kindergarten, and the first few years of elementary school.” Then respondents will be asked to say how much they agree with the following 6 statements: “When I was younger than 10... : “My family usually had enough money for things when I was growing up”, “I grew up in a relatively wealthy neighborhood”, “I felt relatively wealthy compared to the other kids in my school”, “things were often chaotic in my house”, “people often moved in and out of my house on a pretty random basis”, and “I had a hard time knowing what my parent(s) or other people in my house were going to say or do from day-to-day.” Responses are made on 7-point scales ranging from 1: strongly disagree, to 7: strongly agree. Items 9 to 11 inform about the respondents’ socioeconomic status at the time of the inclusion in the experimental protocol. They are taken from the well-validated work by Griskevicius et al. (2011) and ask respondents to say how much they agree with the following 3 statements: “I have enough money to buy things I want”, “I don't need to worry too much about paying my bills”, and “I don't think I'll have to worry about money too much in the future.” Finally, items 12 and 13 inform about the respondents’ health state. More specifically, item 12 asks the question “How is your health in general?”, and proposes 4 responses from 1-“Bad” to 4-“Excellent”. Item 13 asks the respondents to answer –

by choosing a value between 0 and 100 – the question “How much effort do you make to look after your health and ensure your safety these days?”

Right after the questionnaire, the respondents are shown the following scenario:

You have just been presented with 13 additional questions about your personal life. If you agree, we would like to share your answers to these extra questions and link them with the data collected during the past 10 days. We would like to ask you to give your consent for this extra data to be linked to the survey data and shared with other researchers. All data protection regulations will be strictly observed during the process, i.e., the results will be anonymous and will not allow any conclusions to be drawn about your person. Your consent is of course voluntary. You can also revoke it at any time. Do you agree to the extra data being linked and shared with other researchers?

Respondents can either select “yes” or “no” to this last question. Only if respondents select “yes”, will their extra data be linked to the survey data and shared online on the Open Science Framework. We expect that the average behavior in the Trust game on the first day is not or poorly correlated with data sharing, but social trust questionnaire responses and the average behavior in the Trust game over all days are significantly and more strongly correlated with data sharing.”

Please note that the data collected during the entire experiment including the Trust games and the social trust questionnaires will be uploaded to OSF as is standard practice in open science. And only the extra data (i.e., 13 additional questions) from participants who agree to their data being shared will be uploaded to OSF, not from participants who did not agree to their data being shared.

Scientific validity of the research question(s)

This study addresses a topical and important research question, and I only have three (relatively minor) comments in this regard:

1) The authors suggest three "advantages" of economic games on p. 3. Yet, as will become clear just a couple of paragraphs below, at least some of these "advantages" may rather be disadvantages. For instance, to the extent that games indeed capture "actual behavior" ("advantage 1") and assuming that this behavior is highly variable, this is evidently a disadvantage provided that one aims to study a trait. Moreover, exactly due to the lack of reliability as introduced shortly afterwards, games may in fact not be very well suited to "precisely investigate inter-individual differences" ("advantage 3"). These advantages should thus either be introduced more thoroughly and convincingly, or potentially be omitted altogether.

The reviewer is right. While economic games obviously have several advantages, for the purpose of this study, mentioning them does not provide any significant value. Therefore, we chose to omit them from our revised manuscript.

2) Generally the authors argue that games may be more susceptible to state influences whereas self-reports may rather capture a stable trait. Although recent evidence indeed suggests a series of cognitive explanations for the latter hypothesis (e.g., Steiner et al., 2021), some questionnaires may actually be designed to specifically capture states, too (e.g., the state-trait anxiety inventory was developed to assess situational and general dispositions; Spielberger, 2010).

The reviewer is right and we now include this nuance on p.4 and p.5 of the revised manuscript.

Page 4 and 5: “A possible explanation for the instability of lab-based experimental tasks such as economic games, is that decisions in the lab are likely influenced by the current situation (i.e., people’s current states) where the participant is in, while many self-reported questionnaires are specifically designed to capture people’s average behavior across a long period of time (i.e., people’s stable traits) (Palminteri & Chevallier, 2018). Some questionnaires aim to measure participants’ states or both their current state and their stable traits (e.g., the State-trait anxiety questionnaire, Spielberger, 2010). In the specific case we are interested in, social trust questionnaires are designed to measure an individual’s prototypical tendency to trust other people. For example, questionnaire items are often phrased as: “Generally speaking...”. Overall, such design features are geared to reduce the influence of momentary trends (Fleeson, 2004; Wennerhold & Friese, 2020).”

In the case of our study, existing questionnaires on social trust generally focus on generalized social trust (Nannestad, 2008), which is trust in unknown individuals and is often conceptualized as a psychological stable trait. It is classically measured with the following question: “Generally speaking, would you say that most people can be trusted, or that you can’t be too careful in dealing with people”. However, self-reported questionnaires have their own limitations (e.g., social desirability bias). Experimental measures such as the Trust game, which we intend to use, also have the potential to capture stable trust propensity levels. Nevertheless, as we argue in our manuscript, this can be achieved if the sources of noise that can affect an individual’s trust behavior measured at a given moment are controlled for. We think that by repeating the Trust game over an extended period of time could be an efficient way of minimizing the influence of noise on measuring trust propensity and approximating its stability with an experimental procedure.

3) Relatedly, the authors appear to presuppose that social trust is a stable personality disposition. But is trust not highly variable across different contexts, as a function of different personal interactions, etc.? In other words, is the expected variability in trust (as measured by the repeated games; hypothesis 1) simply considered random noise, or is this the result of a systematic mechanism? Overall a more detailed elaboration on how the authors conceptualize social trust would be helpful.

Social trust is usually conceptualized along two dimensions: trust propensity displayed by the trustor and trustworthiness displayed by the trustee (Mayer et al., 1995). In our manuscript, we focus on people’s trust propensity, and when we mention “social trust” throughout the manuscript, we are specifically talking about the trustors’ “trust propensity”. It is indeed true that trust propensity levels vary considerably between countries (e.g., Albanese & de Blasio, 2014; Bjørnskov, 2007) and individuals (e.g., Alesina & La Ferrara, 2002; Guillou et al., 2020; Mell et al., 2021). However, trust propensity within individuals is typically conceptualized as a relatively stable trait (Colquitt et al., 2007; Lyu & Ferrin, 2018; Mayer et al., 1995; Uslaner, 2008). For example, Mayer et al. (1995) argue that trust propensity is ““a trait that is stable across situations”. There is indeed evidence that suggests that trust propensity is consistent across a range of interpersonal relationships (Colquitt et al., 2007; Uslaner, 2008). However, recent evidence suggests that there are also momentary, within-person fluctuations that can affect trust propensity.

Furthermore, there are different types of trust (e.g., in family, in close friends, in institutions, in unknown people generally, etc.). Each type may have different psychological underpinnings and may be sensitive to different factors. Here we are particularly interested in generalized social trust (Nannestad, 2008), which is trust in unknown individuals and is often conceptualized as a psychological stable trait. It is classically measured with the following question: “Generally speaking, would you say that most people can be trusted, or that you can’t be too careful in dealing with people”. The Trust game aims at mimicking this general type of trust with unknown individuals.

The longitudinal nature of our research design, where participants play repeated Trust games on multiple days, should counteract momentary, within-person fluctuations in trust propensity levels.

We have added the aforementioned details on p.5 and on p.6 of the revised manuscript.

Page 5 and 6: “To test this hypothesis, we focus on people’s social trust. Social trust is usually conceptualized along two dimensions: trust propensity displayed by the trustor and trustworthiness displayed by the trustee (Mayer et al., 1995). In our study, social trust refers to the trustors’ propensity to trust others. Trust propensity varies considerably between countries (e.g., Albanese & de Blasio, 2014; Bjørnskov, 2007) and individuals (e.g., Alesina & La Ferrara, 2002; Guillou et al., 2020; Mell et al., 2021). However, an individual’s trust propensity is typically conceptualized as a stable trait (Colquitt et al., 2007; Lyu & Ferrin, 2018; Mayer et al., 1995; Uslaner, 2008). For instance, Mayer et al. (1995) write that trust propensity is “a trait that is stable across situations”. Furthermore, there are different types of trust (e.g., in family, in close friends, in institutions, in unknown people generally, etc.). Each type may have different psychological underpinnings and may be sensitive to different factors. Here we are particularly interested in generalized social trust (Nannestad, 2008), which is trust in unknown individuals. We chose to focus on social trust because the economic game (i.e., the Trust game, see Berg et al., 1995; Johnson & Mislin, 2011) and questionnaire (Glaeser et al., 2000) that are frequently used to measure it are each well-established tools that are standardly used in the literature. Both of these measures try to capture people’s generalized social trust.”

Logic, rationale, and plausibility of the proposed hypotheses

The hypotheses are plausible and clearly specified.

Soundness and feasibility of the methodology and analysis pipeline (including statistical power analysis where applicable)

Generally the empirical approach is sound, but I have a series of comments in this regard.

1) The exclusion criteria concerning "obvious random responses" should be specified more clearly.

We thank the reviewer for highlighting this issue. We removed this exclusion criterion (based in part on a comment from Reviewer #1).

2) Participant recruitment will be stopped "once the target sample is reached"; this requires that the specified quality-control / exclusion criteria will have to be monitored on the fly (otherwise the final sample will be small than the aspired sample size). Is this the case?

We understand the reviewer’s question regarding the target sample size. The target sample of 275 respondents already includes the expected attrition and exclusion of respondents based on our exclusion criteria (see section “2.7 Power analysis and sample size estimation” of the manuscript). Therefore, there is no need to monitor the exclusion criteria as the experiment progresses.

Page 16 and 17: “In addition, an independent study (Kothe & Ling, 2019) found that for a longitudinal study (i.e., 12 months) performed on Prolific the attrition rate was around 25%. Because our study is significantly shorter, we expect a lower attrition rate of 10%. Furthermore, in a study with a similar methodology (Mell et al., 2021), 25% of the respondents were excluded from the analysis for not correctly answering the questions of the last of the two comprehension check sections. To compensate for the

expected attrition and exclusion of respondents (35% in total), we will recruit 275 respondents.”

3) Is there any concern for anchoring effects, given that participants fill in the questionnaire in the first session? What about splitting the sample into two groups (a: questionnaires pre & post repeated assessments; b: questionnaire only post repeated assessments)?

We understand the reviewer's concern on this point and agree that it is an interesting suggestion. However, because we want to examine the stability of the social trust questionnaire responses over time for all participants, it is necessary that all participants fill out the questionnaire at both time points. Regarding the possibility that filling out the social trust questionnaire on the first day has an anchoring effect on Trust game decisions on subsequent days, we would like to emphasize that the trust questionnaire consists of three short items, which are fully aligned with the instructions and goals of the Trust game. Even though we cannot entirely rule out the possibility of an anchoring effect, we think it is unlikely that the presentation of these three items at the end of the 1st session will have a strong impact on people's responses in the Trust game two days later. Anchoring effect lasting several days might be more likely for intrusive or emotionally loaded questionnaires (such as mortality salience, which are not presented on the first day anymore, see previous comment about data sharing scenario) but we do not think that our 3-item trust questionnaire exposes us to that risk. After careful consideration of the reviewer's suggestion, we decided to keep the design as is, because it is well suited to measure the temporal stability of trust questionnaire responses.

4) (How) will it be ensured that participants play the games only every second day?

This will be done in two ways. First, we will send an invitation e-mail via Prolific to the participants on every other day at midnight. This will allow participants to take part in the experiment on that specific day. Second, we will set an expiration date and time on Qualtrics, so that participants can only take part in the experiment for a specific day on that specific date. For example, for the first day of the experiment, an invitation e-mail will be sent on 0:00 at the start of that day, and a time limit will be set for 23:59 at the end of that day.

5) I was confused by the statement "In the 8 intermediate days, respondents only play the Trust game 16 times" (p. 8). Why "only"? They also play 16 rounds in the initial session, or do they play more (if so, please clarify)?

This statement refers to the fact that on the first day, participants fill out the baseline social trust questionnaire AND they play the Trust game (which has 16 rounds), whereas during the 8 intermediate days, they play the Trust game but they do not fill out the questionnaire. In order to avoid any confusion, we changed this sentence in the revised manuscript on p.7 and p.8.

Page 7 and 8: “In the 8 intermediate days, respondents play the Trust game 16 times (detailed instructions and comprehension check sections are again included).”

6) The authors plan to use ICCs to quantify the reliability of the economic games. Instead of computing 16 separate ICCs and then averaging them, I wondered whether one could implement a single model that treats the 16 games (or even the 4 x 4 predictors) as fixed effects, to thus directly extract one overall indicator of intraindividual variability?

We completely agree with the reviewer and are thankful for this helpful comment. We will now use the function *rpt* from the R package *rptR*. This function allows for the integration of fixed effects, in our case “stakes” and “reciprocation probability”. This model calculates 1 ICC, precisely in accordance with the reviewers' suggestion. Finally, based on Reviewer #1's

comments, we decided to use a “Poisson” distribution, because our dependent variable willingness-to-play is on a 9-point Likert scale. We have added the aforementioned details on p.14 of the revised manuscript.

Page 14: “To determine the ICC, we will use the function *rpt* from the R package *prrR*. The dependent variable is the average willingness-to-play, stakes and reciprocation probability will be included as fixed effects and respondents’ ID will be included as a grouping factor with a random intercept. Because the dependent variable is on a 9-point Likert scale, we will use a Poisson distribution.”

7) Although the analysis plan is specified quite well, but the authors should spell out explicitly under which conditions (thresholds for statistical tests, etc.) they consider their hypotheses confirmed.

We thank the reviewer for highlighting this issue and we apologize for these omissions. For hypothesis 1 concerning the variability in economic games decisions, the statistical threshold is already mentioned (i.e., ICC value <0.50 would support our hypothesis). For hypothesis 2 concerning the stability of questionnaire responses, we will use the standard statistical significance threshold of $p < 0.05$. For hypothesis 3 concerning the mixed effects model, the standard threshold of $p < 0.05$ will also be used. Furthermore, we will consider this hypothesis as validated if the effect size is close to 0.20. We have added the aforementioned details on p.14 and on p.15 of the revised manuscript

Page 14: “We will use the standard statistical significance threshold of $p < 0.05$ ”

Page 15: “We will use the standard statistical significance threshold of $p < 0.05$. A correlation slope difference of 0.20 will be taken as the effect size of reference to consider the alternative hypothesis as validated.”

Whether the clarity and degree of methodological detail would be sufficient to replicate exactly the proposed experimental procedures and analysis pipeline
Yes, but see points above.

Whether the authors provide a sufficiently clear and detailed description of the methods to prevent undisclosed flexibility in the experimental procedures or analysis pipeline
Yes, but see points above.

Whether the authors have considered sufficient outcome-neutral conditions (e.g. positive controls) for ensuring that the results obtained are able to test the stated hypotheses
Yes, but there is some ambiguity as to how “close” is “close enough” compared to the results of Mell et al. (p. 11); please specify.

In our Trust game, participants will have to indicate their “willingness-to-play”; this is based on a new methodology developed by Mell et al. (2021). Therefore, we have to base our expected average willingness-to-play on one study making it difficult to give a precise estimation. The reason for this is that there might be substantial variation between studies measuring this average willingness-to-play that we are unaware of. However, because we will use a very similar methodology and the same online platform to recruit our participants, we propose that an acceptable range would be between 4.62 and 7.44 (one standard deviation above and below the average willingness-to-play found by Mell et al., 2021).

We have added the aforementioned details on p.13 of the revised manuscript

Page 13: “To determine if our sample is representative in terms of the average willingness-to-play in the Trust games on the first day, our data will be compared to Mell et al. (2021) who used a similar methodology and found an average value of 6.02 ($SD = 1.49$) in their initial study and 6.03 ($SD = 1.33$) in a replication study. We expect that our mean is close to the means found by Mell et al. (2021), specifically between 4.62 and 7.44, because this is exactly one standard deviation below and above the mean found by Mell et al. (2021).”

References

- Ajzen, I. (2005). Attitudes, personality, and behavior. McGraw-Hill Education (UK).
- Albanese, G., & de Blasio, G. (2014). Who trusts others more? A cross-European study. *Empirica*, 41(4), 803-820. <https://doi.org/10.1007/s10663-013-9238-7>
- Alesina, A., & La Ferrara, E. (2002). Who trusts others?. *Journal of public economics*, 85(2), 207-234. [https://doi.org/10.1016/S0047-2727\(01\)00084-6](https://doi.org/10.1016/S0047-2727(01)00084-6)
- Bjørnskov, C. (2007). Determinants of generalized trust: A cross-country comparison. *Public choice*, 130(1-2), 1-21. <https://doi.org/10.1007/s11127-006-9069-1>
- Brosig, J., Riechmann, T., & Weimann, J. (2007). Selfish in the end?: An investigation of consistency and stability of individual behavior. Unpublished manuscript.
- Chuang, Y., & Schechter, L. (2015). Stability of experimental and survey measures of risk, time, and social preferences: A review and some new results. *Journal of Development Economics*, 117, 151–170. <https://doi.org/10.1016/j.jdeveco.2015.07.008>
- Colquitt, J. A., Scott, B. A., & LePine, J. A. (2007). Trust, trustworthiness, and trust propensity: A meta-analytic test of their unique relationships with risk taking and job performance. *Journal of applied psychology*, 92(4), 909. <https://doi.org/10.1037/0021-9010.92.4.909>
- Frey, R. (2020). Decisions from experience: Competitive search and choice in kind and wicked environments. *Judgment and Decision Making*, 15(2), 282–303.
- Frey, R., Pedroni, A., Mata, R., Rieskamp, J., & Hertwig, R. (2017). Risk preference shares the psychometric structure of major psychological traits. *Science Advances*, 3(10), e1701381. <https://doi.org/10.1126/sciadv.1701381>
- Galizzi, M. M., & Navarro-Martinez, D. (2019). On the external validity of social preference games: a systematic lab-field study. *Management Science*, 65(3), 976-1002. <https://doi.org/10.1287/mnsc.2017.2908>
- Gelman, A., Carlin, J. B., Stern, H. S., Dunson, D. B., Vehtari, A., & Rubin, D. B. (2013). *Bayesian data analysis*. CRC press.
- Gervais, M. M. (2017). RICH economic games for networked relationships and communities: development and preliminary validation in Yasawa, Fiji. *Field methods*, 29(2), 113-129. <https://doi.org/10.1177/1525822X16643709>
- Goeschl, T., Kettner, S. E., Lohse, J., & Schwieler, C. (2015). What do we learn from public good games about voluntary climate action? Evidence from an artefactual field experiment. <http://dx.doi.org/10.2139/ssrn.2620229>
- Guillou, L., Grandin, A., & Chevallier, C. (2020). Correcting misperceptions of relative income: Impact on temporal discounting and social trust. *PsyArXiv*. <https://doi.org/10.31234/osf.io/vwyfn>
- Henrich, J., Boyd, R., Bowles, S., Camerer, C., Fehr, E., Gintis, H., ... & Tracer, D. (2005). "Economic man" in cross-cultural perspective: Behavioral experiments in 15 small-scale societies. *Behavioral and brain sciences*, 28(6), 795-855. <http://dx.doi.org/10.1017/S0140525X05000142>

- Henrich, J., Heine, S. J., & Norenzayan, A. (2010). The weirdest people in the world?. *Behavioral and brain sciences*, 33(2-3), 61-83. <https://doi.org/10.1017/S0140525X0999152X>
- Hertwig, R., & Ortmann, A. (2001). Experimental practices in economics: A methodological challenge for psychologists? *Behavioral and Brain Sciences*, 24(03), 383–403.
- Jacquet, P. O., Safra, L., Wyart, V., Baumard, N., & Chevallier, C. (2019). The ecological roots of human susceptibility to social influence: a pre-registered study investigating the impact of early-life adversity. *Royal Society open science*, 6(1), 180454. <https://doi.org/10.1098/rsos.180454>
- Johnson, D.R., & Creech, J.C. (1983). Ordinal measures in multiple indicator models: A simulation study of categorization error. *American Sociological Review*, 48, 398-407. <https://doi.org/10.2307/2095231>
- Kolar, D. W., Funder, D. C., & Colvin, C. R. (1996). Comparing the accuracy of personality judgments by the self and knowledgeable others. *Journal of personality*, 64(2), 311-337. <https://doi.org/10.1111/j.1467-6494.1996.tb00513.x>
- Koster, J. M., & Leckie, G. (2014). Food sharing networks in lowland Nicaragua: an application of the social relations model to count data. *Social Networks*, 38, 100-110. <https://doi.org/10.1016/j.socnet.2014.02.002>
- Krumpal, I. (2013). Determinants of social desirability bias in sensitive surveys: a literature review. *Quality & Quantity*, 47(4), 2025-2047. <https://doi.org/10.1007/s11135-011-9640-9>
- Lettinga, N., Mell, H., Algan, Y., Jacquet, P.O., & Chevallier, C. (2021). Childhood environmental adversity is not linked to lower levels of cooperative behaviour in economic games. *Evolutionary Human Sciences*, 3, 1-22. <https://doi.org/10.1017/ehs.2021.21>
- Lönnqvist, J.-E., Verkasalo, M., Walkowitz, G., & Wichardt, P. C. (2015). Measuring individual risk attitudes in the lab: Task or ask? An empirical comparison. *Journal of Economic Behavior & Organization*, 119, 254–266. <https://doi.org/10.1016/j.jebo.2015.08.003>
- Lyu, S. C., & Ferrin, D. L. (2018). Determinants, consequences, and functions of interpersonal trust: What is the empirical evidence?. <https://doi.org/10.4324/9781315745572-7>
- Mayer, R. C., Davis, J. H., & Schoorman, F. D. (1995). An integrative model of organizational trust. *Academy of Management Review*, 20(3), 709–734. <https://doi.org/10.5465/amr.1995.9508080335>
- McAuliffe, W. H., Forster, D. E., Pedersen, E. J., McCullough, M. E., & Baumert, A. (2019). Does cooperation in the laboratory reflect the operation of a broad trait?. *European Journal of Personality*, 33(1), 89-103. <https://doi.org/10.1002/per.2180>
- McElreath, R. (2020). *Statistical rethinking: A Bayesian course with examples in R and Stan*. CRC press.
- Mell, H., Safra, L., Demange, P., Algan, Y., Baumard, N., & Chevallier, C. (2021). Early life

adversity is associated with diminished social trust in adults. *Psyarxiv*.
<https://doi.org/10.31234/osf.io/43q8z>

- Montgomery, J. M., Nyhan, B., & Torres, M. (2018). How conditioning on posttreatment variables can ruin your experiment and what to do about it. *American Journal of Political Science*, 62(3), 760-775. <https://doi.org/10.1111/ajps.12357>
- Nannestad, P. (2008). What have we learned about generalized trust, if anything?. *Annu. Rev. Polit. Sci.*, 11, 413-436.
<https://doi.org/10.1146/annurev.polisci.11.060606.135412>
- Norman, G. (2010). Likert scales, levels of measurement and the "laws" of statistics. *Advances in Health Sciences Education*, 15(5), 625-632. <https://doi.org/10.1007/s10459-010-9222-y>
- Pisor, A. C., Gervais, M. M., Purzycki, B. G., & Ross, C. T. (2020). Preferences and constraints: the value of economic games for studying human behaviour. *Royal Society open science*, 7(6), 192090. <https://doi.org/10.1098/rsos.192090>
- Pisor, A., & Ross, C. T. (2021). How generalizable are patterns of parochial altruism in humans?.
- Rozin, P. (2009). What kind of empirical research should we publish, fund, and reward?: A different perspective. *Perspectives on Psychological Science*, 4(4), 435-439.
<https://doi.org/10.1111/j.1745-6924.2009.01151.x>
- Safra, L., Chevallier, C., & Palminteri, S. (2019). Depressive symptoms are associated with blunted reward learning in social contexts. *PLoS computational biology*, 15(7), e1007224. <https://doi.org/10.1371/journal.pcbi.1007224>
- Schniter, E., Shields, T. W., & Sznycer, D. (2020). Trust in humans and robots: Economically similar but emotionally different. *Journal of Economic Psychology*, 78, 102253.
<https://doi.org/10.1016/j.joep.2020.102253>
- Spielberger, C. D. (2010). State-Trait anxiety inventory. The Corsini encyclopedia of psychology. <https://doi.org/10.1002/9780470479216.corpsy0943>
- Steiner, M., D., Seitz, F., & Frey, R. (in press). Through the window of my mind: Mapping information integration and the cognitive representations underlying self-reported risk preference. *Decision*. <https://doi.org/10.31234/osf.io/sa834>
- Sullivan, G. M., & Artino Jr, A. R. (2013). Analyzing and interpreting data from Likert-type scales. *Journal of graduate medical education*, 5(4), 541.
<https://doi.org/10.4300/JGME-5-4-18>
- Tiokhin, L., Hackman, J., Munira, S., Jesmin, K., & Hruschka, D. (2019). Generalizability is not optional: insights from a cross-cultural study of social discounting. *Royal Society open science*, 6(2), 181386. <https://doi.org/10.1098/rsos.181386>
- Uslaner, E. M. (2008). Where you stand depends upon where your grandparents sat: The inheritability of generalized trust. *Public opinion quarterly*, 72(4), 725-740.
<https://doi.org/10.1093/poq/nfn058>
- Zumbo, B. D., & Zimmerman, D. W. (1993). Is the selection of statistical methods governed

by level of measurement?. *Canadian Psychology*, 34(4), 390.
<https://doi.org/10.1037/h0078865>

Appendix C

Subject: Resubmitted Manuscript RSOS-210213.R2

Dear Pr. Chambers,

Thank you for your helpful review of our manuscript and for providing us with the opportunity to revise and resubmit our work. We are grateful for the in principle acceptance of our registered report, subject to one minor change. We wish to thank the reviewer for his/her helpful comment. We provide a reply to the reviewer's comment, with changes to the manuscript located below.

We hope you will find this version of the manuscript acceptable.

With kind regards,

Niels Lettinga, Lou Safra, Pierre O. Jacquet, Coralie Chevallier

Editor

One of the previous Stage 1 reviewers was available to assess the revised manuscript. The assessment is positive, and there is just one minor methodological point remaining to address. Provided the authors can respond adequately to this concern through revision or rebuttal, Stage 1 in-principle acceptance should be forthcoming without requiring further in-depth review.

Reviewer 2

Comment

The authors have done a great job in revising their registered report. I do not see any major outstanding issues and only have a minor re-comment. Specifically, the authors write:

"Because the dependent variable is on a 9-point Likert scale, we will use a Poisson distribution." Although a Poisson distribution is potentially better suited as compared to the Gaussian dist., it still is not quite the right distribution for modeling Likert responses (i.e., unlike the Likert scales, Poisson distributions do not have an upper bound). The appropriate model would be an ordered logistic model and the authors may want to consider using it -- which, however, makes it substantially more intricate to interpret the respective estimates.

I wish the authors good luck with the data collection and look forward to reading the final manuscript.

We agree with the reviewer that an ordered logistic model would be appropriate in this case. We will now fit an ordered logistic regression model using the function *clmm2* from the R package *Ordinal*. After that, we will calculate the ICC using the following formula:

$$\text{ICC} = \tau / (\tau + (\pi^2))$$

where τ is the between group variance, in our case the estimated value of the random factor parameter "respondents' ID" (Liu, 2015)

We have added the aforementioned details on p.14 of the revised manuscript.

Page 14: "To test the hypothesis that decisions in the Trust game show substantial variability over time, we will measure the test-retest reliability of the willingness-to-play via the intraclass correlation coefficient (ICC), which is a widely used reliability index. To determine the ICC, we will first fit an ordinal logistic regression model using the function *clmm2* from the R package *Ordinal*. The dependent variable is willingness-to-play, stakes and reciprocation probability will be included as fixed effects and respondents' ID will be included as a grouping factor with a random intercept. After that, we will calculate the ICC using the following formula: $\text{ICC} = \tau / (\tau + (\pi^2))$ where τ is the between group variance, in our case the estimated value of the random factor parameter "respondents' ID" (Liu, 2015). Values <0.50 indicate poor reliability, between 0.50 and 0.75 moderate reliability, between 0.75 and 0.90 good reliability and >0.90 excellent reliability. Because we expect that the Trust game decisions show substantial variability over time, values <0.50 would support our hypothesis."

References

Liu, X. (2015). *Applied Ordinal Logistic Regression Using Stata: From Single-Level to Multilevel Modeling*. SAGE Publications.

Appendix D

Dear Prof. Chambers,

We have now completed the second stage of our pre-registered report. As we committed to, we have conducted our study using the exact experimental procedures and analysis plan described in the pre-registered report. All the data was collected after we received the in-principle acceptance. The experiments have been executed and analysed in the manner originally approved. The positive controls were validated, and the data quality checks confirmed the quality of our data. Links to the repository of the Stage 1 Registered Report and of the data and analyses scripts are provided on page 18 of the manuscript.

Our analyses confirmed the instability of participants' behaviour in the Trust Games over the sessions and the stability of their responses to the social trust questionnaire. However, contrary to our hypothesis, we found a significant and similarly sized association between participants' responses to the social trust questionnaire and their behaviour in the Trust Games, whether this behaviour was measured only in the first session or averaged across sessions. Finally, pre-registered exploratory analysis of changes in participants' behaviour in the Trust Games over time revealed the existence of different profiles in our sample of participants.

We explain these results at the end of the abstract as well as in a Materials and Methods subsection and in a Results section, added to the Stage 1 Registered Report. We have also added a short 'Unregistered analysis' paragraph in which we present complementary analyses conducted to ensure that our null results on real-life cooperative behaviour were not due to bias in our measures. We conclude our manuscript by discussing our results and how economic games can be used in conjunction with questionnaires in the study of social trust. Finally, compared to the Stage 1 Registered Report, we have replaced 'Days' with 'Sessions' to make the presentation of our study clearer and we have changed the American spelling of 'behavior' to the English spelling 'behaviour'.

We thank you in advance for considering the second stage of our pre-registered report and look forward to your feedback.

Sincerely,

Lou Safra, Niels Lettinga, Pierre O. Jacquet and Coralie Chevallier